# MULTI-AGENT-AS-JUDGE: ALIGNING LLM-AGENT-BASED AUTOMATED EVALUATION WITH MULTI-DIMENSIONAL HUMAN EVALUATION

## ABSTRACT

Nearly all human work is collaborative; thus, the evaluation of real-world NLP applications often requires multiple dimensions that align with diverse human perspectives. As real human evaluator resources are often scarce and costly, the emerging "LLM-as-a-judge" paradigm sheds light on a promising approach to leverage LLM agents to believably simulate human evaluators. Yet, to date, existing LLM-as-a-judge approaches face two limitations: persona descriptions of agents are often arbitrarily designed, and the frameworks are not generalizable to other tasks. To address these challenges, we propose **MAJ-EVAL**, a **M**ulti-**A**gent-as-**J**udge evaluation framework that can automatically construct multiple evaluator personas with distinct dimensions from relevant text documents (e.g., research papers), instantiate LLM agents with the personas, and engage in-group debates with multi-agents to generate multi-dimensional feedback. Our evaluation experiments in both the educational and medical domains demonstrate that MAJ-EVAL can generate evaluation results that better align with human experts' ratings compared with conventional automated evaluation metrics and existing LLM-as-a-judge methods.

## 1 INTRODUCTION

Nearly all human work today is multi-person collaboration work Olson & Olson (2000). As a result, the evaluation of NLP applications in the real world, such as those in education and healthcare, often requires considering multiple dimensions that align with diverse human perspectives He et al. (2023); Chen et al. (2024). In particular, such evaluations cannot be handled by a single evaluator or traditional similarity-based metrics (e.g., ROUGE-L) because complex real-world scenarios and the collaborative nature of human work necessitate integrating insights from diverse stakeholders who bring different domain-specific roles and perspectives to the evaluation Liu et al. (2024). For instance, care providers, family caregivers, and patients have different needs while evaluating patient summaries generated by LLMs Yang et al. (2025); similarly, assessing LLM-generated Question-Answering pairs for children's reading comprehension requires feedback from children, parents, and teachers Chen et al. (2025b).

While human expert annotation remains the gold standard for such domain-specific real-world evaluations, collecting multi-dimensional human feedback is highly challenging due to the expert resource scarcity and the tremendous cost (time and money) for recruitment Yao (2024); Lu et al. (2023). To address these challenges, recent work has explored the use of Large Language Models (LLMs) as evaluators to substitute human evaluators, giving rise to the "LLM-as-a-judge" paradigm Zheng et al. (2023); Zhuge et al. (2024). In particular, the multi-agent framework under this paradigm employs multiple role-playing LLM agents to simulate human evaluation Park et al. (2024); Ran et al. (2025); Lu et al. (2025), where each agent is intended to reflect one human evaluative dimension Kim et al. (2024); Li et al. (2024c). Such a framework offers a promising approach for evaluating real-world applications where multi-dimensional human feedback is required.

Despite its promise, the current multi-agent evaluation approach faces two critical limitations: First, the **design of agent personas is often arbitrary and not generalizable** due to the lack of a systematic methodology Li et al. (2024b). For example, even within the same task, studies may focus on

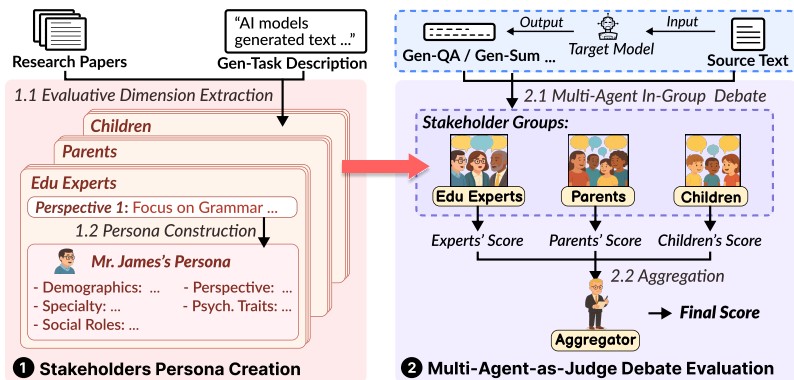

Figure 1: The overview of MAJ-EVAL's two-step design: Step 1, MAJ-EVAL extracts stakeholders' diverse perspectives from the provided research papers and constructs corresponding personas for the LLM agents. Step 2, agents within the same stakeholder group engage in an in-group debate. An aggregator agent synthesizes ratings from all groups to derive a final score.

different dimensions due to varied priorities and interpretation: one study may handcraft a "teacher" agent that focuses on "grammar accuracy," while another might handcraft the same agent prioritizing "student engagement," leading to results that cannot be reliably reproduced across studies or by other research teams Park et al. (2023); Li et al. (2025). Second, **most evaluation setups are not easily adaptable** because they are specifically designed for a particular task or scenario. For instance, an evaluation pipeline designed for medical summarization may include dimensions like "clinical consistency," but these are irrelevant for similar summarization tasks for children's education, where "child engagement" is more appropriate. Because these dimensions and role definitions are hard-coded per task Lin & Chen (2023), the evaluation frameworks often require complete redesigns to handle new domains, undermining scalability and transferability.

To overcome these limitations, we propose **MAJ-EVAL**: a **M**ulti-**A**gent-as-**J**udge evaluation framework that automates the construction and deployment of human-aligned LLM agents for robust real-world natural language generation (NLG) evaluation. As illustrated in Figure 1, MAJ-EVAL first identifies descriptions about different human stakeholders and their perspectives (e.g., teachers emphasizing educational value) from researcher-provided documents related to the domain-specific tasks (e.g., research papers). These descriptions are then transformed into structured agent personas attributes, such as domain expertise, psychological traits, and social roles. Second, the resulting LLM multi-agents engage in an in-group debate to reflect, challenge, and refine their initial judgments before producing an aggregated rating aligned with multi-dimensional human ratings.

We evaluate MAJ-EVAL on two challenging domain-specific real-world tasks: (1) question-answer generation (QAG) for children's storybook reading Xu et al. (2022) and (2) multi-document summarization of medical literature DeYoung et al. (2021). Across both tasks, MAJ-EVAL achieves more substantial alignment with human ratings compared to traditional automated metrics (e.g., ROUGE-L Lin (2004), BERTScore Zhang et al. (2019)), single LLM-as-a-judge evaluation (e.g., G-Eval Liu et al. (2023)), and existing multi-agent approaches (e.g., ChatEval Chan et al. (2023)). These results underscore the value of grounding evaluation in human-aligned multi-dimensional evaluation and demonstrate MAJ-EVAL's potential as a generalizable framework for real-world NLG evaluation.

## 2 RELATED WORK

### 2.1 TRADITIONAL EVALUATION METHODS FOR NLG

Automated evaluation metrics have long been the standard for measuring the performance of NLG systems, primarily due to their simplicity and scalability Yao et al. (2023b). Metrics such as ROUGE Lin (2004), BLEU Papineni et al. (2002), and BERTScore Zhang et al. (2019) are widely adopted in both research and industry settings. These methods typically compute the token-level or embedding-based similarity between model outputs and a set of reference texts. However, such

Figure 2: MAJ-EVAL's multi-agent debate evaluation process (Step 2). Agents first provide independent evaluations. Then, the moderator coordinates a free-form debate, allowing agents to discuss and refine their evaluation. Finally, the aggregator synthesizes all evaluations into a final evaluation.

similarity-based metrics often fall short in real-world, domain-specific tasks where deeper contextual understanding, factual correctness, and task-specific appropriateness are more critical than surface-level overlap Yao et al. (2023c); Sai et al. (2022); Zhu et al. (2023); Wu et al. (2025). For example, in medical summarization generation Croxford et al. (2024), ROUGE may fail to penalize hallucinated claims that are fluently expressed but unsupported by evidence. Similarly, in educational QA generation Zhao et al. (2022); Chen et al. (2025b), lexical similarity cannot assess whether a question is pedagogically meaningful for children.

To address the limitations of automated metrics, human evaluation has become the gold standard for assessing generated text, especially in domain-specific tasks Chen et al. (2024); Croxford et al. (2024). Human expert annotators are typically asked to rate outputs across multiple dimensions (e.g., fluency, relevance, educational value, or clinical accuracy) Lu et al. (2023). While comprehensive, human evaluation is costly, labor-intensive, and often lacks consistency across different research projects Belz et al. (2021); Yao et al. (2023a); Thomson et al. (2024). Moreover, the complexity and human workflows of many real-world applications mean that no single human annotator can represent the relevant multi-stakeholder perspectives. For instance, in evaluating interactive storybook content for children Xu et al. (2021); Chen et al. (2025b), each stakeholder may provide different evaluation dimensions even for the same generated content from the same model: a teacher may prioritize educational value, while a parent may focus on emotional engagement. This diversity is both necessary and difficult to scale using traditional human evaluation protocols.

## 2.2 LLM-AS-A-JUDGE EVALUATION

Researchers have proposed leveraging LLMs as evaluators, which is commonly referred to as the "LLM-as-a-judge" evaluation paradigm Zheng et al. (2023). In this setup, a single LLM is prompted or fine-tuned to assess the model-generated text, simulating human evaluation criteria such as relevance, coherence, and correctness Li et al. (2024a); Lee et al. (2024); Fu et al. (2023). Representative methods include G-Eval Liu et al. (2023), which guides GPT-4 using chain-of-thought prompting for structured dimension-wise assessment, and PandaLM Wang et al. (2023b), which fine-tunes an LLaMA-7B model for preference ranking. These methods are lightweight and scalable but inherit several challenges. Most notably, they reflect **single-model bias**, where judgments are constrained by the model's own training data and reasoning style, thus may fail to simulate multi-stakeholder perspectives in real-world evaluations Yao et al. (2024).

To mitigate the limitations of single-LLM evaluation, recent work has extended the paradigm to multi-agent setups, where multiple LLM agents, each adopting a distinct persona or evaluative role, collaborate or debate to arrive at a final assessment Chen et al. (2023); Zhu et al. (2023). Examples include ChatEval Chan et al. (2023), which assigns agents to pre-defined roles such as "general public" or "critic," and MADISSE Koupaee et al. (2025), which frames evaluation as a debate between agents with opposing initial stances. These systems improve diversity in judgment and better mirror real-world evaluative complexity. However, most of these approaches still rely on manually crafted personas and predefined evaluation dimensions, limiting reproducibility and cross-task generalization Szymanski et al. (2025); Gebreegziabher et al. (2025). For example, an agent labeled as

a "critic" in one task may not exhibit the same evaluative priorities in another, and a dimension like "factual consistency" may not translate well from summarization to dialogue generation.

## 3 MAJ-EVAL

We propose MAJ-EVAL, an LLM-based multi-agent evaluation framework designed to simulate real-world multi-stakeholder-aligned NLG evaluation. As shown in Figure 1, MAJ-EVAL enables researchers to evaluate model-generated content by (1) automatically extracting stakeholder perspectives from domain-specific documents and constructing diverse agent personas grounded in those perspectives, and (2) orchestrating in-group debates among these agents to produce final, multi-dimensional evaluation scores.

### 3.1 STAKEHOLDER PERSONA CREATION

The first stage of MAJ-EVAL focuses on creating personas that faithfully represent the diverse evaluative dimensions found in real-world stakeholder groups. To ensure both coverage and credibility, persona creation follows a two-step process: (1) extracting evaluative dimensions from research publications, and (2) constructing personas based on those extracted perspectives.

***Step 1: Evaluative Dimension Extraction.*** Given a list of documents of domain-specific tasks (e.g., research papers) $L = \{l_1, \ldots, l_n\}$, MAJ-EVAL uses an LLM $M_\theta$ to identify relevant **stakeholders** and extract their associated perspectives (i.e., evaluative **dimensions**). Each document is parsed to locate stakeholders (e.g., "parents," "clinicians") and their descriptive attributes (e.g., priorities, values), along with evidence-based evaluation dimensions (e.g., "focus on grammar correctness"). The output for each document $l_i$ is a structured list of stakeholder tuples $s_{ij} = (n_{ij}, c_{ij}, \mathcal{V}_{ij})$, where $n_{ij}$ denotes the stakeholder's name, $c_{ij}$ is their description, and $\mathcal{V}_{ij}$ is a set of (dimension, evidence) pairs. For instance, in the task of QAG for children's story reading, one extracted evaluative dimension of the parents is "Parents expect questions to stimulate creativity, critical thinking, and curiosity rather than factual recall...", with the evidence of "The majority of participants felt that current AI tools were 'silly'..." from a paper that explores parents' expectations and perceptions of AI-assisted reading tools for children Sun et al. (2024).

To unify overlapping roles and ensure coherent persona design, MAJ-EVAL aggregates similar stakeholders into groups using semantic clustering via LLM $M_\theta$. Within each group, redundant or semantically close dimensions are automatically merged, resulting in a consolidated view of each stakeholder group. For example, education technology developers who emphasize "system usability" and AI developers who promote "system robustness" are grouped under a "system developer" stakeholder group with multiple evaluative dimensions. Following prior work showing that diverse perspectives can enhance the debate process Liang et al. (2024), MAJ-EVAL retains distinct evaluative dimensions within each group to preserve diversity. More examples of extracted stakeholders' (dimension, evidence) pairs for the children's book QAG task are shown in Table 10, and examples for the medical summarization generation task are shown in Table 11 in Appendix A.9. Descriptions of each task are showin in Appendix A.11. The prompt for this step is shown in Table 12.

***Step 2: Dimension-Based Persona Construction.*** For each consolidated dimension within a stakeholder group, MAJ-EVAL constructs a detailed persona: $p_{ij} = M_\theta(c_i, v_{ij}, e_{ij})$. Inspired by prior work on LLM-based role-play agents Chen et al. (2025a), each persona includes five key attributes: (1) **demographic information** (e.g., name, age, profession), (2) **evaluative dimension** (from earlier perspective extraction), (3) **domain specialty**, (4) **psychological traits**, and (5) **social relationships**. These personas serve as the basis for instantiating stakeholder-aligned agents during evaluation. We include examples of constructed personas in Table 6 and the corresponding prompt in Table 13. In addition, Appendix A.8 presents an example of MAJ-EVAL's complete persona creation workflow.

### 3.2 MULTI-AGENT-AS-JUDGE DEBATE EVALUATION

In the second stage of MAJ-EVAL, the constructed personas are instantiated as LLM-based agents that engage in a **multi-agent-as-judge debate evaluation** (Table 14 presents the instantiation prompt). Each stakeholder group (e.g., teachers, clinicians) evaluates model-generated outputs through in-group deliberation (in-group multi-agent free debate), simulating how real-world stake-

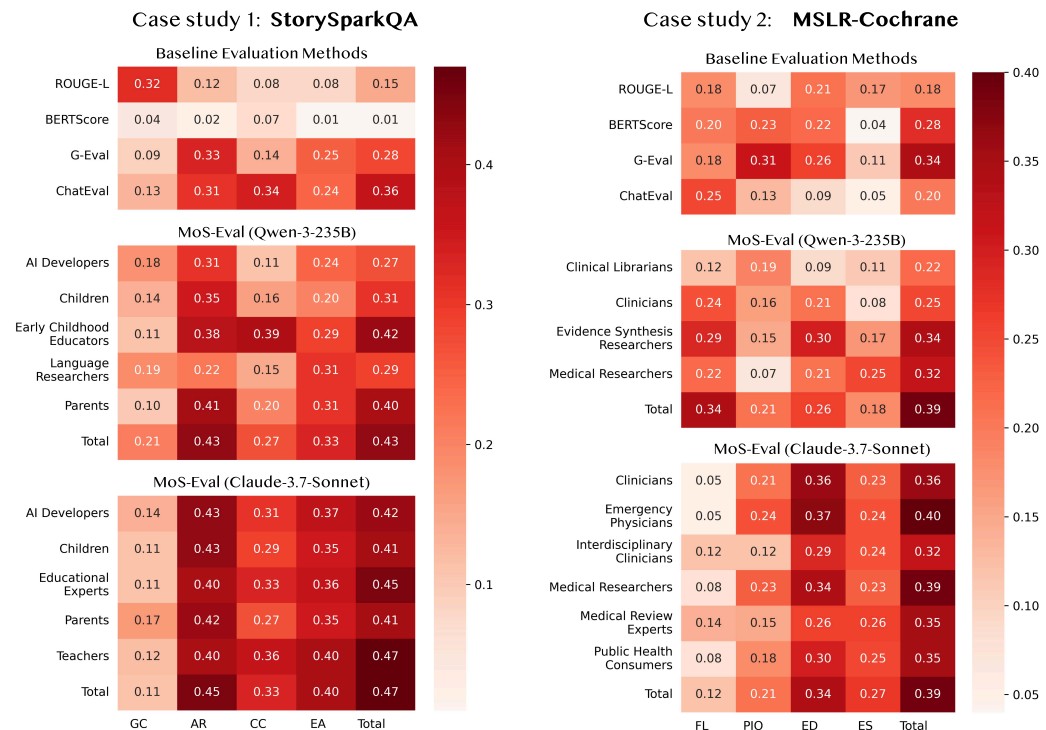

Figure 3: Spearman correlations between evaluation methods and human ratings on `StorySparkQA` (left) and MSLR-COCHRANE (right). **Higher values indicate stronger human alignment.** `StorySparkQA`'s dimensions include Grammar Correctness (GC), Answer Relevancy (AR), Contextual Relevancy (CC), and Educational Appropriateness (EA). MSLR-COCHRANE includes Fluency (FL), PIO Consistency (PIO), Effect Direction (ED), and Evidence Strength (ES).

holders might discuss, disagree, and eventually converge on evaluation judgments. The debate process is divided into three phases: (1) individual agent-as-a-judge evaluation, (2) multi-agent in-group free debate, and (3) aggregation of scores into a final group judgment (see Figure 2 and Algorithm 1).

***Phase 1: Individual Agent-as-a-Judge.*** Each stakeholder agent begins by independently assessing the generated output according to their unique perspective and expertise. This phase aims to capture a diversity of opinions, reflecting how different stakeholders may initially interpret the same content in task-specific ways. The prompt for this phase is presented in Table15.

***Phase 2: Multi-Agent In-Group Free Debate.*** Next, the agents engage in an open-ended multi-turn debate within each group. Moderated by a coordinating agent, the debate unfolds dynamically, prioritizing agents with unresolved disagreements or unaddressed perspectives. Agents challenge, reflect on, or reinforce each other's views and revise their evaluations as needed. This phase encourages surfacing blind spots, resolving conflicts, and generating more refined judgments. We include the prompt for phase 2 in Table 16.

***Phase 3: Aggregation.*** Finally, an aggregator agent aggregates the updated evaluations across all agent groups in two ways: (1) synthesizing the qualitative feedback from all stakeholder agents' final evaluations and (2) computing an average score of each group's post-debate quantitative ratings. Table 17 shows the prompt for this phase.

# 4 EXPERIMENTAL SETUP

## 4.1 TASKS AND DATASETS

Our first evaluation task is a Narrative Question-Answer Generation (QAG) task from children's storybooks, we utilize the **StorySparkQA** dataset Chen et al. (2024), an expert-annotated QA

dataset designed for 3- to 6-year-old children's interactive story-reading activity. `StorySparkQA` consists of 5,868 QA pairs that are derived from children's fairytale stories and enriched with real-world knowledge. In our experiment, we evaluate the 70 QA pairs generated by GPT-4 as reported by Chen et al. (2024) that have been annotated by human experts using the following four evaluation dimensions: *Grammar Correctness* (i.e., whether the QA pair is grammatically correct), *Answer Relevancy* (i.e., whether the answer meaningfully addresses the question), *Contextual Consistency* (i.e., whether the QA pair is grounded in the story but introduces external real-world knowledge), and *Children's Educational Appropriateness* (i.e., whether the QA pair is suitable for 3- to 6-year-old children in the context of story-reading).

Our second evaluation task is a multi-document summary generation for medical literature reviews. We choose the **MSLR-COCHRANE** dataset Wang et al. (2023a), an expert-annotated benchmark comprising 600 model-generated summaries from six models. These summaries were annotated by domain experts along four dimensions: *Fluency* (i.e., whether the summary is fluent in English), *PIO Consistency* (i.e., whether the Population, Intervention, and Outcome (PIO) align with the target summary), *Effect Direction* (the reported impact of the intervention), and *Evidence Strength* (the degree to which the claim is supported by the underlying studies). For our study, we construct a representative evaluation set by randomly sampling 17 summaries from each of the six models, resulting in a balanced subset of 102 generated summaries.

## 4.2 BASELINE EVALUATION METHODS

We compare MAJ-EVAL against the following three types of evaluation methods as baselines:

**Single Metrics of Automated Evaluation.** We adopt two commonly used similarity-based automated metrics to evaluate generated outputs. **ROUGE-L F1** Lin (2004) measures surface-level similarity by computing lexical overlap with reference texts. Since lexical overlap may fail to capture deeper semantic meaning, we also use **BERTScore** Zhang et al. (2019), which measures semantic similarity using contextual embeddings from pre-trained language models.

**Single LLM-as-a-judge Evaluation** We use **G-EVAL** Liu et al. (2023), a prompting-based evaluation framework that guides a single LLM to rate the generated content along specific dimensions. We experiment with both GPT-4 OpenAI (2023), Claude-3.7-Sonnet, and Qwen-3-235B as the base models for G-EVAL evaluations.

**Multi-Agent-as-Judge Evaluation** We adopt **ChatEval** Chan et al. (2023), a multi-agent evaluation framework where agents role-play different personas to assess generated outputs. We experiment with both GPT-4, Claude-3.7-Sonnet, and Qwen-3-235B as the underlying models and follow the default setup to assign personas.

For both single-LLM and multi-agent evaluation methods, we follow the original prompt structure proposed by the authors, with slight modifications to adapt it to our two case study tasks. Full prompt details are provided in Appendix A.10.

## 4.3 EVALUATION METRICS

In order to evaluate how well each evaluation method aligns with the multi-dimensional human judgments in each dataset, we report the absolute **Spearman's rank correlation coefficient** ($\rho$), following Chan et al. (2023); Chu et al. (2024).

In addition, we report **Kendall's Tau** ($\tau$) Sen (1968), which assesses the ordinal ranking consistency between the models' scores and human ratings, and **Pearson's correlation coefficient** Cohen et al. (2009), which calculates linear relationships between the models' scores and human ratings (refer to Appendix A.4). To assess the internal consistency of agent evaluations during in-group debates, we compute inter-coder reliability within each stakeholder group using **Krippendorff's Alpha** Krippendorff (2011), shown in Appendix A.7.

## 4.4 IMPLEMENTATION DETAILS

We experimented with Claude-3.7-Sonnet and Qwen3-235B Yang et al. (2024) as the underlying models of MAJ-EVAL. For Qwen, we employed the sglang Zheng et al. (2024) inference framework

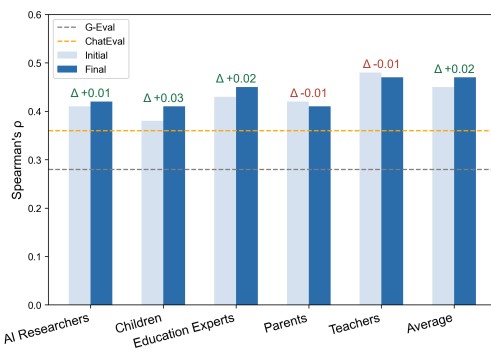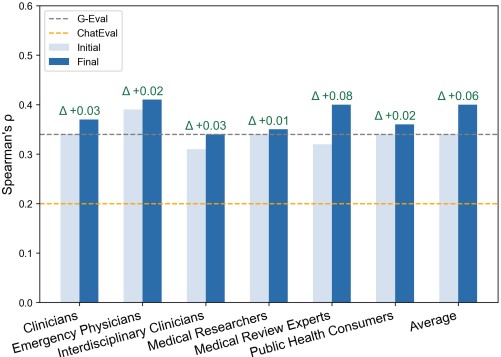

Figure 4: Spearman correlations of MAJ-EVAL (Claude-3.7-Sonnet) stakeholder agents' initial and final scores with human ratings on `StorySparkQA` (left) and MSLR-COCHRANE (right). **Dark blue bars higher than light blue bars indicate improved human alignment after debate.**

with the temperature set to 0.6. For Claude, we accessed the model via the AWS API, using its default temperature setting of 1.0.

We employ Google Scholar as our search engine and follow the snowballing search strategy Wohlin (2014) for input document selection. Specifically, we (1) locate the qualitative research publications cited in each dataset paper (if there is no citation to qualitative research, we use keywords of the NLG task description), and (2) conduct an additional keyword-based search combining task-relevant keywords with "qualitative interview" on Google Scholar to retrieve recent publications from the past three years, ensuring broader coverage and up-to-date perspectives. From the results, we selected three representative documents for the QAG task in children's story-reading Xu et al. (2021); Chen et al. (2025b); Sun et al. (2024) and two documents for the medical summarization task Yang et al. (2023); Yun et al. (2023) based on recency, task relevance, and resource considerations.

## 5 EXPERIMENT RESULTS

### 5.1 OVERALL EVALUATION PERFORMANCE

Our results show that similarity-based metrics, including ROUGE-L and BERTScore, exhibited overall weak correlations with human ratings on most evaluation dimensions across both `StorySparkQA` and MSLR-COCHRANE (see Figure 3). It is notable that ROUGE-L achieves its highest correlation on the *Grammar Correctness* dimension, but falls short in capturing deeper domain-specific dimensions such as *Educational Appropriateness* or *Effect Direction*. We attribute this discrepancy to ROUGE-L's focus on token-level similarity.

For the LLM-as-a-judge methods, G-Eval outperformed similarity-based metrics across both datasets, showing better human alignment on most domain-specific dimensions. ChatEval, which integrates multiple LLM agents, showed improvements on `StorySparkQA` but did not yield gains on MSLR-COCHRANE. These findings suggest that LLM-as-a-judge methods can achieve closer alignment with human evaluations when carefully designed. However, their effectiveness depends on manually crafted prompts and does not readily generalize across different tasks and domains.

Worth noting that our MAJ-EVAL outperformed all baseline methods across both tasks and the majority of evaluation dimensions. When used with different underlying models, MAJ-EVAL demonstrated consistent, robust alignment with human ratings, particularly on domain-specific dimensions. Detailed computational and time costs of MAJ-EVAL are reported in Appendix A.5.

### 5.2 DOMAIN-SPECIFIC DIMENSIONS ALIGNMENT

Across both domains, baseline methods exhibited inconsistent performance. For instance, ROUGE-L showed a higher correlation with `StorySparkQA`'s *Grammar Correctness* but lower with MSLR-COCHRANE's *Fluency*. BERTScore underperformed on `StorySparkQA` but excelled on MSLR-COCHRANE. Similarly, G-Eval and ChatEval varied in alignment with human ratings across

different tasks. We also observe that these methods struggled to align well with humans on domain-specific dimensions like *Educational Appropriateness* for the children's QAG task and *Effect Direction* for the medical summarization task.

We attribute this inconsistency to different task settings. The QAG task in children's story-reading requires the integration of external real-world knowledge beyond the source story. Therefore, model-generated QA pairs often include knowledge content divergent from human-authored content, reducing the effectiveness of similarity-based metrics. In contrast, MSLR-COCHRANE's datapoints include generated and reference summary pairs that are both grounded in the same source documents, making metrics like BERTScore more effective for capturing textual similarity.

Remarkably, MAJ-EVAL consistently exhibited superior alignment with human ratings across both domains, particularly on domain-specific dimensions. In `StorySparkQA`, stakeholder groups such as Teachers and Educational Experts from the Claude-3.7-Sonnet variant and Early Childhood Educators from the Qwen-3-235B variant showed the highest correlations on *Context Relevancy* and *Children's Educational Appropriateness*. This strong alignment is consistent with their real-world domain familiarity with both pedagogical goals and children's cognitive needs Cade (2023). However, MAJ-EVAL showed a weaker correlation on *Grammar Correctness*. This discrepancy may arise because the stakeholder agents tend to prioritize educational and developmental appropriateness over strict grammatical accuracy.

For MSLR-COCHRANE, MAJ-EVAL also correlated highly with humans, particularly on the *Effect Direction* and *Evidence Strength*. Among the stakeholder agents, Emergency Physicians and Clinicians in the Claude-3.7-Sonnet variant showed strong alignment with human ratings on *Effect Direction*, which is critical for interpreting intervention outcomes. Medical Researchers in both variants performed best on *Evidence Strength*, which reflects the certainty and quality of clinical summaries. These results exhibit the effectiveness of MAJ-EVAL's stakeholder-grounded personas in capturing domain-specific evaluative dimensions. While MAJ-EVAL achieved reasonable correlations on *PIO Consistency*, it lagged behind BERTScore and G-Eval. We believe it is because similarity-based metrics are better at capturing how well the generated summary matches the reference on specific document-anchored components, such as population, intervention, and outcome.

We include additional comparison and qualitative analysis of the evaluation outcomes of G-Eval, ChatEval, and MAJ-EVAL, alongside human ratings in Appendix A.6.

## 5.3 ABLATION STUDY

### 5.3.1 EFFECTIVENESS OF MAJ-EVAL'S PERSONA CREATION

To evaluate the effectiveness of MAJ-EVAL's persona creation step, we conduct an ablation study by assigning each stakeholder with a simple role definition (e.g., "You are a preschool teacher who often reads books to your students." for the teacher agent) instead of the detailed persona. We then compute the Spearman correlation ($\rho$) between each group's scores and human ratings under both the simple role and detailed persona conditions. The results are presented in Appendix A.3.

Across both domains, MAJ-EVAL correlates more closely with human ratings on both the overall quality and individual evaluation dimensions. This observation justifies that our proposed implementation of MAJ-EVAL (i.e., a detailed persona construction process and a debate mechanism) led to evaluation metrics that correlated higher with human ratings.

### 5.3.2 IMPACT OF MAJ-EVAL'S MULTI-AGENT IN-GROUP FREE DEBATE MECHANISM

To examine the impact of the in-group debate mechanism, we extract each stakeholder agent's initial score before the in-group debate and final score post the debate (see Section 3.2) and calculate the correlation (Spearman's $\rho$) between each group's scores and human ratings before and after the debate. The results are presented in Figure 4 and Figure 5 in Appendix A.4.

In both domains, we observe that many stakeholder groups' initial evaluations already exhibit strong alignment with human ratings, demonstrating the effectiveness of MAJ-EVAL's persona construction step. Importantly, all task-level averages increased after the debate, with 15 out of 20 stakeholder groups showing positive gains. This improvement suggests that the in-group debate mechanism effectively supports most stakeholder agents in refining their evaluations.

However, a few groups, like the Language Researchers for the children's QAG task, exhibited reduced correlations after the debate. Our analysis of their debate logs reveals that while their initial scores adhered closely to the task description, the debating process led these stakeholders to consider additional evaluative dimensions, such as inferential scaffolding and vocabulary richness. Although these dimensions extend beyond those used in human ratings, these extended dimensions reflect theoretical concerns rooted in early childhood education Vygotsky (1978); Wasik et al. (2006). Thus, we believe the in-group debate step can further enrich human evaluation with more comprehensive evaluative dimensions specific to the task.

## 6 DISCUSSION

Comparing MAJ-EVAL with automated metrics, single- and multi-LLM evaluation methods, we observe that MAJ-EVAL consistently achieves higher correlations with human ratings on domain-specific dimensions (e.g., *Educational Appropriateness*). However, a trade-off emerges between MAJ-EVAL's performance on domain-specific and textual-level dimensions (e.g., *Grammar Correctness*), indicating that stakeholder agents, shaped by their perspectives, tend to prioritize domain-specific dimensions over surface-level linguistic fidelity. This tendency aligns with real-world human evaluation behaviors Clark et al. (2021).

These findings highlight the importance of aligning evaluation methods with task-specific objectives. We recommend applying MAJ-EVAL in evaluating NLP applications that involve diverse user needs, multiple social roles, or domain expertise. For evaluating surface-level linguistic fidelity, traditional automated metrics or single LLM-as-a-judge methods may remain more suitable.

Overall, across both tasks, MAJ-EVAL exhibits consistently stronger correlations with human ratings on task-specific dimensions. This observation justifies the cross-domain generalizability of MAJ-EVAL as well as the effectiveness of integrating multi-stakeholder evaluative dimensions in evaluating generated text for real-world tasks.

## 7 CONCLUSION

In summary, this work presents MAJ-EVAL, a multi-agent evaluation framework designed for real-world NLG evaluation. MAJ-EVAL 1) derives stakeholders' evaluative dimensions from domain-specific documents, 2) constructs stakeholder agent personas grounded in these dimensions, and 3) organizes LLM agents into in-group debates to collaboratively generate evaluations. Our case studies on two domain-specific tasks, namely QAG in children's interactive story-reading and medical summarization, demonstrate that MAJ-EVAL's stakeholder-grounded evaluations achieve stronger alignment with human ratings on multiple domain-specific dimensions compared with existing automated metrics, single-LLM evaluations, and multi-agent evaluation methods.

## 8 LIMITATIONS

This work focuses on the design and development of a multi-agent evaluation framework tailored to real-world text generation scenarios. While our case studies highlight the effectiveness of MAJ-EVAL compared to existing automated metrics and both single- and multi-LLM evaluation methods, several limitations remain. First, although our case studies in children's interactive QA and medical summarization show promising results, these domains represent only a subset of real-world applications with limited human ratings. Future work could examine additional domains to further assess the framework's generalizability. Second, there remain limited datasets annotated by a diverse set of stakeholders. We call for future work on collecting data that reflects multiple stakeholder perspectives. Third, as the baseline methods (i.e., G-Eval and ChatEval) are backed by LLMs with large parameters, we did not test MAJ-EVAL with a wider range of models(e.g., Llama 3 Touvron et al. (2023)). In the future, we can test MAJ-EVAL using smaller models to better understand the framework's compatibility across model scales.

## 9 REPRODUCIBILITY STATEMENT

To support reproducibility, we provide MAJ-EVAL's source code with detailed instructions for installation, configuration, and execution in the supplementary material. The anonymized version of our source code is also accessible at `https://anonymous.4open.science/r/MAJ-Eval-51BA/`. The source code includes scripts for persona creation, debate evaluation, and analysis, along with sample data, generated personas, and human annotation scores. Model configurations and parameters are explicitly documented, and all experimental setups, datasets, and evaluation metrics are described in the main text and appendix. We hope these resources will support the research community in replicating and extending our work.

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

# A APPENDIX

## A.1 THE USE OF LARGE LANGUAGE MODELS (LLMS)

We employed LLMs solely as a general-purpose assistive tool for grammar checking and refinement of word choices to enhance clarity. LLMs' involvement was limited to improving the readability and fluency of sentences, without contributing to research ideation, analysis, or substantive writing.

## A.2 IN-GROUP DEBATE ALGORITHM

Algorithm 1 illustrate the MAJ-EVAL's in-group debate process.

---

**Algorithm 1** In-Group Multi-Agent Debate

---

**Require:** Task description $T$, Evaluation content $C$, Evaluation format $F$
    **Optional:** Related context $X$
1: Initialize number of stakeholders $n$; max debate rounds $m$
2: Initialize coordinator agent $Coo$
3: Initialize aggregator agent $Agg$
4: **for** $i = 1$ to $n$ **do**
5:     Instantiate stakeholder agent $A_i$
6: **end for**

---

    **Phase 1: Independent Evaluation**
7: **for** $i = 1$ to $n$ **do**
8:     $E_i \leftarrow A_i.\text{Evaluate}(T, C, F, X)$
9: **end for**

---

    **Phase 2: Free Debate**
10: Initialize $FinishedAgents \leftarrow \emptyset$
11: $H \leftarrow \{E_1, \ldots, E_n\}$; $FinalFeedback \leftarrow \emptyset$
12: **for** $t = 1$ to $m$ **do**
13:     $S_t \leftarrow Coo.\text{SelectNextSpeaker}(H)$
14:     $F_t \leftarrow S_t.\text{Respond}(T, C, F, X, H)$
15:     Append $F_t$ to $H$
16:     **if** $F_t$ contains "NO MORE COMMENTS" and $S_t \notin FinishedAgents$ **then**
17:         Append $F_t$ to $FinalFeedback$
18:         Add $S_t$ to $FinishedAgents$
19:     **end if**
20:     **if** $|FinalAgents| = n$ **then**
21:         **break**
22:     **end if**
23: **end for**

---

    **Phase 3: Final Aggregation**
24: $Result \leftarrow Agg.\text{Aggregate}(H, FinalFeedback)$

---

## A.3 ABLATION STUDY RESULTS WITH SIMPLE ROLE DEFINITION

We conducted an ablation study by assigning each stakeholder a straightforward and simple role description (e.g., prompt: "You are a preschool teacher who often reads books to your students"). The Spearman correlation analysis results for the StorySparkQA and MSLR-COCHRANE datasets are shown in Table 1 and 2.

## A.4 COMPLETE EXPERIMENTAL RESULTS

We present the complete performance of MAJ-EVAL along with all the baselines on the QAG in children's story-reading and medical summarization task in Table 3 and 4, respectively. In addition,

| `StorySparkQA` | Overall Quality | Grammar Correctness | Answer Relevancy | Contextual Consistency | Educational Appropriateness |
|---|---|---|---|---|---|
| Simple-Role Qwen-3-235B | 0.42 | 0.12 | 0.43 | 0.27 | 0.31 |
| MoS-Eval Qwen-3-235B | 0.43 | **0.18** | 0.43 | 0.27 | 0.33 |
| Simple-Role Claude-3.7-Sonnet | 0.43 | 0.09 | **0.45** | 0.28 | 0.37 |
| MoS-Eval Claude-3.7-Sonnet | **0.47** | 0.14 | **0.45** | **0.33** | **0.40** |

Table 1: Spearman's $\rho$ correlations between different evaluation methods and human ratings for `StorySparkQA`. Higher values indicate stronger alignment with human judgments. **Bolded** numbers are the overall best scores.

| **MSLR-CochrANE** | Overall Quality | Fluency | PIO Consistency | Effect Directions | Evidence Strength |
|---|---|---|---|---|---|
| Simple-Role Qwen-3-235B | 0.24 | 0.26 | 0.14 | 0.15 | 0.12 |
| MoS-Eval Qwen-3-235B | 0.39 | **0.34** | **0.21** | 0.26 | 0.18 |
| Simple-Role Claude-3.7-Sonnet | 0.30 | 0.08 | 0.16 | 0.30 | 0.18 |
| MoS-Eval Claude-3.7-Sonnet | **0.40** | 0.10 | **0.21** | **0.34** | **0.28** |

Table 2: Spearman's $\rho$ correlations between different evaluation methods and human ratings for `StorySparkQA`. Higher values indicate stronger alignment with human judgments. **Bolded** numbers are the overall best scores.

| `StorySparkQA` | Overall Quality | | | Grammar Correctness | | | Answer Relevancy | | | Contextual Consistency | | | Educational Appropriateness | | |
|---|---|---|---|---|---|---|---|---|---|---|---|---|---|---|---|
| | $r$ | $\rho$ | $\tau$ | $r$ | $\rho$ | $\tau$ | $r$ | $\rho$ | $\tau$ | $r$ | $\rho$ | $\tau$ | $r$ | $\rho$ | $\tau$ |
| Rouge-L | 0.20 | 0.15 | 0.11 | **0.24** | **0.32** | **0.25** | 0.13 | 0.12 | 0.09 | 0.10 | -0.08 | -0.06 | 0.14 | 0.08 | 0.06 |
| BERTScore | 0.11 | 0.01 | 0.01 | 0.04 | -0.04 | -0.02 | 0.10 | 0.02 | 0.01 | 0.06 | -0.07 | -0.05 | 0.09 | -0.01 | -0.01 |
| G-Eval (GPT4) | 0.28 | 0.28 | 0.20 | 0.09 | 0.09 | 0.08 | 0.33 | 0.33 | 0.25 | -0.00 | 0.14 | 0.10 | 0.27 | 0.25 | 0.19 |
| G-Eval (Qwen-3-235B) | 0.35 | 0.26 | 0.18 | 0.18 | 0.16 | 0.13 | 0.36 | 0.29 | 0.21 | 0.09 | 0.11 | 0.08 | 0.31 | 0.20 | 0.15 |
| G-Eval (Claude-3.7-Sonnet) | 0.45 | 0.31 | 0.23 | 0.16 | 0.20 | 0.15 | 0.44 | 0.39 | 0.29 | 0.19 | 0.19 | 0.14 | 0.36 | 0.20 | 0.16 |
| ChatEval | 0.45 | 0.36 | 0.29 | 0.06 | 0.13 | 0.12 | 0.34 | 0.31 | 0.26 | **0.32** | **0.34** | **0.31** | 0.41 | 0.24 | 0.20 |
| ChatEval (Qwen-3-235B) | 0.24 | 0.21 | 0.15 | 0.20 | 0.24 | 0.21 | 0.19 | 0.16 | 0.12 | 0.09 | 0.04 | 0.03 | 0.21 | 0.20 | 0.17 |
| ChatEval (Claude-3.7-Sonnet) | 0.26 | 0.20 | 0.15 | 0.20 | 0.24 | 0.21 | 0.25 | 0.19 | 0.15 | 0.12 | 0.08 | 0.06 | 0.16 | 0.16 | 0.13 |
| MAJ-EVAL Qwen-3-235B | 0.52 | 0.43 | 0.32 | 0.15 | 0.18 | 0.14 | 0.43 | 0.43 | 0.31 | 0.30 | 0.27 | 0.21 | **0.45** | 0.33 | 0.24 |
| *AI Developers* | 0.35 | 0.27 | 0.20 | 0.14 | 0.14 | 0.11 | 0.33 | 0.31 | 0.23 | 0.10 | 0.11 | 0.10 | 0.33 | 0.24 | 0.18 |
| *Children* | 0.38 | 0.31 | 0.24 | 0.09 | 0.11 | 0.09 | 0.34 | 0.35 | 0.27 | 0.22 | 0.16 | 0.12 | 0.30 | 0.20 | 0.15 |
| *Early Childhood Educators* | 0.48 | 0.42 | 0.31 | 0.17 | 0.19 | 0.16 | 0.40 | 0.38 | 0.29 | 0.29 | 0.39 | 0.31 | 0.39 | 0.29 | 0.22 |
| *Language Researchers* | 0.42 | 0.29 | 0.23 | 0.10 | 0.10 | 0.09 | 0.27 | 0.22 | 0.17 | 0.28 | 0.15 | 0.13 | 0.42 | 0.31 | 0.24 |
| *Parents* | 0.46 | 0.40 | 0.30 | 0.15 | 0.21 | 0.16 | 0.41 | 0.41 | 0.31 | 0.23 | 0.20 | 0.16 | 0.39 | 0.31 | 0.23 |
| MAJ-EVAL Claude-3.7-Sonnet | **0.53** | **0.47** | **0.35** | 0.06 | 0.14 | 0.11 | **0.48** | **0.45** | **0.33** | 0.31 | 0.33 | 0.26 | **0.45** | **0.40** | **0.30** |
| *AI Developers* | 0.47 | 0.42 | 0.33 | 0.02 | 0.11 | 0.09 | 0.44 | 0.43 | 0.34 | 0.25 | 0.31 | 0.25 | 0.41 | 0.37 | 0.31 |
| *Children* | 0.48 | 0.41 | 0.31 | 0.03 | 0.11 | 0.09 | 0.48 | 0.43 | 0.32 | 0.23 | 0.29 | 0.23 | 0.41 | 0.35 | 0.28 |
| *Educational Experts* | 0.50 | 0.45 | 0.34 | 0.13 | 0.17 | 0.13 | 0.41 | 0.40 | 0.30 | 0.32 | 0.33 | 0.27 | 0.42 | 0.36 | 0.28 |
| *Parents* | 0.46 | 0.41 | 0.31 | 0.03 | 0.12 | 0.10 | 0.41 | 0.42 | 0.32 | 0.27 | 0.27 | 0.22 | 0.39 | 0.35 | 0.27 |
| *Teachers* | 0.54 | 0.47 | 0.37 | 0.08 | 0.11 | 0.09 | 0.44 | 0.40 | 0.32 | 0.36 | 0.36 | 0.29 | 0.45 | 0.40 | 0.33 |

Table 3: Pearson correlation coefficient ($r$), Spearman's $\rho$ correlations, and Kendall's $\tau$ between evaluation methods and human ratings for `StorySparkQA`. Higher values indicate stronger alignment with human judgments. **Bolded** numbers are the overall best scores. Underlined numbers are the second best scores.

Figure 5 presents Spearman's correlation between MAJ-EVAL (Qwen-3-235B) stakeholder agents' initial (pre-debate) and final (post-debate) scores and human judgments.

## A.5 COMPUTATIONAL COST OF MAJ-EVAL

Based on our records, the average token consumption for the Stakeholder Persona Creation stage is about 34,103 tokens per document, depending on document length. During the debate stage, each stakeholder group uses approximately 18,281 tokens per datapoint. If personas are generated from two documents and there are debates in four stakeholder groups, the total token usage per task is roughly 141,329 tokens. At Claude 3.7 Sonnet's pricing of \$3 per million tokens, the cost is roughly \$0.42 per task. For Qwen-3-235B, which is open-source, the token cost would be even lower.

| MSLR-COCHRANE | Overall Quality | | | Fluency | | | PIO Consistency | | | Effect Direction | | | Evidence Strength | | |
|---|---|---|---|---|---|---|---|---|---|---|---|---|---|---|---|
| | $r$ | $\rho$ | $\tau$ | $r$ | $\rho$ | $\tau$ | $r$ | $\rho$ | $\tau$ | $r$ | $\rho$ | $\tau$ | $r$ | $\rho$ | $\tau$ |
| Rouge-L | 0.18 | 0.18 | 0.13 | -0.2 | -0.18 | -0.15 | 0.07 | 0.07 | 0.05 | 0.25 | 0.21 | 0.18 | 0.16 | 0.17 | 0.13 |
| BERTScore | 0.22 | 0.28 | 0.19 | 0.21 | 0.20 | 0.16 | 0.23 | 0.23 | 0.17 | 0.18 | 0.22 | 0.18 | -0.02 | 0.04 | 0.03 |
| G-Eval | 0.35 | 0.34 | 0.24 | 0.18 | 0.18 | 0.15 | **0.33** | **0.31** | **0.23** | 0.25 | 0.26 | 0.22 | 0.11 | 0.11 | 0.08 |
| G-Eval (Claude-3.7-Sonnet) | 0.40 | 0.34 | 0.24 | 0.04 | 0.02 | 0.02 | 0.22 | 0.22 | 0.17 | **0.37** | 0.32 | 0.27 | 0.26 | 0.21 | 0.16 |
| G-Eval (Qwen-3-235B) | 0.25 | 0.20 | 0.14 | 0.07 | 0.03 | 0.02 | 0.14 | 0.29 | 0.21 | 0.25 | 0.17 | 0.14 | 0.09 | 0.02 | 0.02 |
| ChatEval | 0.18 | 0.20 | 0.16 | **0.28** | 0.25 | 0.23 | 0.08 | 0.13 | 0.11 | 0.09 | 0.09 | 0.09 | 0.07 | 0.05 | 0.05 |
| ChatEval (Claude-3.7-Sonnet) | 0.25 | 0.21 | 0.16 | 0.04 | 0.05 | 0.04 | 0.17 | 0.15 | 0.13 | 0.23 | 0.21 | 0.19 | 0.12 | 0.08 | 0.07 |
| ChatEval (Qwen-3-235B) | 0.28 | 0.34 | 0.27 | 0.13 | 0.11 | 0.10 | 0.17 | 0.14 | 0.11 | 0.18 | 0.23 | 0.21 | 0.21 | **0.28** | **0.24** |
| MAJ-EVAL Qwen3-235B | 0.38 | **0.39** | 0.27 | 0.27 | **0.34** | **0.28** | 0.19 | 0.21 | 0.15 | 0.27 | 0.26 | 0.22 | 0.22 | 0.18 | 0.14 |
| *Clinical Librarians* | 0.20 | 0.22 | 0.16 | 0.14 | 0.12 | 0.10 | 0.16 | 0.19 | 0.15 | 0.09 | 0.09 | 0.08 | 0.13 | 0.11 | 0.09 |
| *Clinicians* | 0.26 | 0.25 | 0.18 | 0.18 | 0.24 | 0.21 | 0.19 | 0.16 | 0.13 | 0.20 | 0.21 | 0.19 | 0.08 | 0.08 | 0.06 |
| *Evidence Synthesis Researchers* | 0.35 | 0.34 | 0.26 | 0.26 | 0.29 | 0.26 | 0.14 | 0.15 | 0.11 | 0.29 | 0.30 | 0.27 | 0.17 | 0.17 | 0.14 |
| *Medical Researchers* | 0.32 | 0.32 | 0.25 | 0.23 | 0.22 | 0.20 | 0.07 | 0.07 | 0.06 | 0.22 | 0.21 | 0.20 | 0.27 | 0.25 | 0.22 |
| MAJ-EVAL Claude-3.7-Sonnet | **0.42** | **0.39** | **0.29** | 0.11 | 0.12 | 0.10 | 0.22 | 0.21 | 0.16 | 0.35 | **0.34** | **0.29** | **0.28** | 0.27 | 0.21 |
| *Clinicians* | 0.39 | 0.36 | 0.28 | 0.07 | 0.05 | 0.05 | 0.20 | 0.21 | 0.17 | 0.36 | 0.36 | 0.32 | 0.24 | 0.23 | 0.19 |
| *Emergency Room Physicians* | 0.41 | 0.40 | 0.32 | 0.06 | 0.05 | 0.05 | 0.24 | 0.24 | 0.20 | 0.37 | 0.37 | 0.35 | 0.24 | 0.24 | 0.21 |
| *Interdisciplinary Clinicians* | 0.35 | 0.32 | 0.27 | 0.14 | 0.12 | 0.12 | 0.12 | 0.12 | 0.10 | 0.30 | 0.29 | 0.28 | 0.25 | 0.24 | 0.22 |
| *Medical Researchers* | 0.40 | 0.39 | 0.32 | 0.08 | 0.08 | 0.08 | 0.27 | 0.23 | 0.20 | 0.33 | 0.34 | 0.32 | 0.23 | 0.23 | 0.21 |
| *Medical Systematic Review Experts* | 0.37 | 0.35 | 0.28 | 0.14 | 0.14 | 0.13 | 0.18 | 0.15 | 0.12 | 0.27 | 0.26 | 0.25 | 0.28 | 0.26 | 0.24 |
| *Public Health Consumers* | 0.38 | 0.35 | 0.29 | 0.11 | 0.08 | 0.08 | 0.19 | 0.18 | 0.16 | 0.31 | 0.30 | 0.29 | 0.27 | 0.25 | 0.23 |

Table 4: Pearson correlation coefficient ($r$), Spearman's $\rho$ correlations, and Kendall's $\tau$ between evaluation methods and human ratings for MSLR-COCHRANE. Higher values indicate stronger alignment with human judgments. **Bolded** numbers are the overall best scores. Underlined numbers are the second best scores.

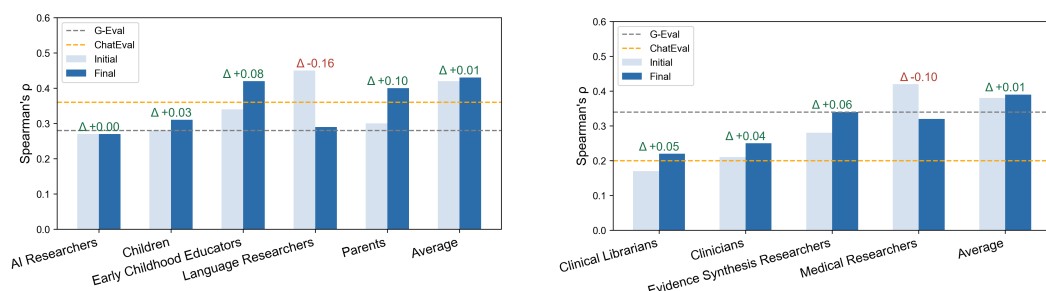

Figure 5: Spearman correlations of MAJ-EVAL (Qwen-3-235B) stakeholder agents' initial and final scores with human judgments on `StorySparkQA` (left) and MSLR-COCHRANE (right). **Dark blue bars higher than light blue bars indicate improved alignment with human ratings after in-group debate.**

Regarding latency, MAJ-EVAL processes a single task in about 26.13 seconds on Qwen-3-235B and 34.20 seconds on Claude 3.7 Sonnet, which is practical for offline evaluation. Moreover, the framework is highly scalable since stakeholder group debates can be executed in parallel.

Overall, the computational cost of MAJ-EVAL is significantly lower than real-world human expert evaluation, which typically requires hours to days for annotation, involves substantially higher budgets, and often faces challenges in recruiting qualified experts.

## A.6 QUALITATIVE ANALYSIS

We randomly sampled one example each from `StorySparkQA` and MSLR-COCHRANE (see Appendix A.6.1 and A.6.2) to qualitatively compare and analyze the evaluation outcomes of G-Eval, ChatEval, and MAJ-EVAL, alongside human ratings. Table 5 presents an overall comparison of the three methods with human annotations. In both evaluations, ChatEval and MAJ-EVAL output qualitative analysis of the QA pair while G-Eval only outputs a score. Within ChatEval, two agent roles (the General Public and the Critic) are designed to simulate collaborative assessment. However, the two agents' feedback struggles to align with the domain-specific dimensions used by humans, primarily reiterating generic task instructions such as integrating real-world knowledge in QA pairs or ensuring clarity in clinical summaries.

| Dataset | Method | Quantitative Score | Qualitative Result | Captured Dimensions |
|---|---|---|---|---|
| StorySparkQA | Human Annotation | 4.56 | - | Grammar Correctness, Answer Relevancy, Contextual Consistency, Educational Appropriateness |
| | G-Eval | 3.00 | No | - |
| | ChatEval | 3.00 | Yes | Contextual Consistency |
| | MAJ-EVAL | 4.33 | Yes | Grammar Correctness, Contextual Consistency, Educational Appropriateness, Engagement |
| MSLR-COCHRANE | Human Annotation | 0.50 | - | Fluency, PIO Consistency, Effect Direction, Evidence Strength |
| | G-Eval | 1.00 | No | - |
| | ChatEval | 0.75 | Yes | Clarity, Domain Relevancy |
| | MAJ-EVAL | 0.50 | Yes | Fluency, Terminology Relevancy, Effect Direction |

Table 5: Qualitative and quantitative comparison of G-Eval, ChatEval, and MAJ-EVAL on the StorySparkQA and MSLR-COCHRANE datasets. On the MSLR-COCHRANE dataset, scores are normalized to 0–1 to match the human rating scale.

As MAJ-EVAL leverages multiple human-aligned evaluative dimensions, its final aggregated evaluations encompass the dimensions used for expert annotations. For instance, in the task of QAG for children's story-reading, the evaluation from the Teacher group highlights dimensions such as contextual relevance (i.e., how well the QA connects the story to real-world knowledge) and educational appropriateness (e.g., simplicity and age suitability of language). These dimensions are derived from the extracted dimensions of the Teacher stakeholder group. For instance, the teacher persona's emphasis on "using simple 'what' questions to inspire children's thinking during story interactions" (Table 6) is reflected in MAJ-EVAL's final evaluation.

In addition, we observe that MAJ-EVAL's agents often introduce evaluative dimensions **beyond** existing human evaluation dimensions. In the QAG task for children's story-reading, the Teacher agents discuss the value of follow-up questions to foster deeper thinking. In the medical summarization task, Medical Researcher agents highlight the significance of clinical specificity. These examples demonstrate that MAJ-EVAL's multi-dimensional evaluation not only align with established human evaluation dimensions but also offer complementary, stakeholder-grounded insights for domain-specific real-world evaluations.

For the final evaluation, we show two snippets of MAJ-EVAL's aggregated evaluation results of one stakeholder group along with the outputs of G-Eval and ChatEval for both the StorySparkQA and MSLR-COCHRANE datasets in Appendix A.6.1 and A.6.2.

### A.6.1 EXAMPLE OF EVALUATION OUTPUTS FOR STORYSPARKQA

**Story Text**: Now the rich neighbor got together as many mowers as ever he could; but the poor one could not hire a single man. At last he despaired altogether and wept, because he did not know how he could manage to get so much as a bit of hay for the cow.
**Question-Answer Pair**: what do you use to cut hay? a mower.

**Averaged Human Score**: 4.56
**G-Eval** Score: 3
**ChatEval** Score: 3

**General Public**: Evaluation evidence: Assistant 1 provided a question-answer pair that was relevant to the story context. However, the question did not introduce new real-world knowledge beyond the story content. Assistant 2 did not provide any question-answer pair, making it impossible to evaluate its performance.
The score of Assistant one: 3
The score of Assistant two: 1

| | |
|---|---|
| | *Children - Emily Thompson* |

**Demographic Info**: A 4-year-old girl from Seattle, living with her parents, attending preschool and developing strong language skills.

**Evaluative Dimension**: Emily is naturally curious and seeks to understand the world through asking countless questions about everything she encounters.

**Specialty**: Demonstrates exceptional verbal curiosity and quick learning ability in exploring new concepts during story interactions.

**Psychological Traits**: Highly inquisitive, easily excited by new information, and displays a playful approach to learning through storytelling and questioning.

**Social Relationships**: Enjoys interactive storytelling with parents and teachers, frequently engaging them with follow-up questions about story content.

*Parent - Rachel Bennett*

**Demographic Info**: A 35-year-old marketing professional and mother of two, balancing work and family responsibilities.

**Evaluative Dimension**: Rachel wants to introduce real-world common knowledge beyond story content during interactive reading sessions.

**Specialty**: Skilled at finding creative ways to expand her children's understanding through contextual learning.

**Psychological Traits**: Pragmatic, goal-oriented, and deeply committed to her children's educational development.

**Social Relationships**: Collaborative with teachers and open to technological tools that support her parenting goals.

*Teacher - Emma Watson*

**Demographic Info**: A 27-year-old early education teacher specializing in interactive learning techniques.

**Evaluative Dimension**: Emma prefers using simple 'what' questions to inspire children's thinking during story interactions.

**Specialty**: Adept at designing age-appropriate questioning strategies that encourage active knowledge acquisition.

**Psychological Traits**: Energetic, intuitive, and passionate about nurturing children's natural curiosity.

**Social Relationships**: Frequently shares teaching insights with colleagues and participates in educational research projects.

*Educational Expert - Dr. Karen Rodriguez*

**Demographic Info**: A 45-year-old professor of early childhood education with extensive research experience.

**Evaluative Dimension**: Karen seeks to create educational content that is age-appropriate and knowledge-expanding for young learners.

**Specialty**: Leading researcher in cognitive development and interactive learning strategies for preschool children.

**Psychological Traits**: Analytical, methodical, and deeply committed to evidence-based educational approaches.

**Social Relationships**: Collaborates with AI developers, teachers, and researchers to advance educational technology.

*AI Developer - Dr. Sophia Martinez*

**Demographic Info**: A 40-year-old computer scientist and AI ethics researcher with a background in educational technology.

**Evaluative Dimension**: Sophia aims to develop AI tools that support open and diverse interactions encouraging creative thinking.

**Specialty**: Expert in creating AI systems that prioritize cognitive stimulation over pure information delivery.

**Psychological Traits**: Principled, forward-thinking, and committed to ethical technological innovation.

**Social Relationships**: Actively engages with multidisciplinary teams to ensure responsible AI development.

Table 6: Examples of constructed stakeholder personas for the QAG task of children's interactive story-reading. We randomly sampled one persona from each stakeholder group.

**Critic**: Evaluation evidence: Assistant 1 provided a question-answer pair that was relevant to the story context. However, the question did not introduce new real-world knowledge beyond the story content. Assistant 2 did not provide any question-answer pair, making it impossible to evaluate its

| *Clinician - Dr. Sarah Thompson* |
|---|
| **Demographic Info**: 39-year-old female with a Medical Doctor (MD) degree and specialization in Internal Medicine. Works in a hospital emergency ward in San Francisco and currently pursuing a masterŽ2019s in medical genetics. |
| **Evaluative Dimension**: Prioritizing evidence from biomedical literature that considers patient-specific characteristics like demographics and comorbidities is essential for validating medical suggestions. |
| **Specialty**: Diagnosing and managing rare genetic disorders combined with expertise in interpreting personalized medicine datasets. |
| **Psychological Traits**: Analytical, cautious, detail-oriented, and empathetic. Prefers decisions backed by reliable data integration. |
| **Social Relationships**: Frequently collaborates with Clinical Librarians for tailored literature searches and Medical Students during patient consultations. |
| *Clinical Librarian - Mr. James Middlebrook* |
| **Demographic Info**: 60-year-old male with a bachelor's in Library Science and extensive experience as a Clinical Librarian in London, UK. |
| **Evaluative Dimension**: Favors integrating gray literature in scenarios where robust evidence is scarce while focusing on applicability over methodological rigor. |
| **Specialty**: Tracking underrepresented studies like conference abstracts and unpublished medical reports for comprehensive analysis. |
| **Psychological Traits**: Adventurous reader, strong problem-solving skills, interested in resourceful literature uncovering, and supportive of interdisciplinary work. |
| **Social Relationships**: Shares gray literature findings with Medical Researchers and consults with Epidemiologists on rural case studies. |
| *Medical Researcher - Ms. Priya Ranganathan* |
| **Demographic Info**: 25-year-old female medical student completing her clinical rotations at a university hospital in Mumbai, India. |
| **Evaluative Dimension**: Appreciates structured summaries that align with clinical training benchmarks and build critical thinking simultaneously. |
| **Specialty**: Leveraging Evidence-Based Medicine (EBM) tools for bedside Case-Based Learning (CBL) and adaptive clinical reasoning. |
| **Psychological Traits**: Hardworking, curious, enthusiastic about digital tools, and goal-oriented in skill development. |
| **Social Relationships**: Collaborates with Clinical Researchers on evidence integration projects and mentors younger students in review techniques. |
| *Evidence Synthesis Researcher - Dr. Thomas Kessler* |
| **Demographic Info**: 61-year-old male retired systematic reviewer currently volunteering for health-oriented AI ethics councils in Munich, Germany. |
| **Evaluative Dimension**: Stresses concerns about LLM-facilitated reviews devaluing rigorous methods and stifling transformative thought. |
| **Specialty**: Critical appraisal of AI in medical research integrity and promoting methodological orthodoxy in hybrid workflows. |
| **Psychological Traits**: Guardian of traditional rigor, skeptical of automation exuberance, philosophically oriented in educational dynamics. |
| **Social Relationships**: Advises Emerging Researchers on writing innovations while collaborating with Editors on AI-review conflict resolution. |

Table 7: Examples of constructed stakeholder personas for the summarization task of multiple medical literature. We randomly sampled one persona from each stakeholder group.

performance.
The score of Assistant one: 3
The score of Assistant two: 1

**MAJ-EVAL's Teacher Group**: 4.33

**Areas of Agreement** All three stakeholders agree that the question-answer pair:

- Successfully connects to the storybook content about mowers.
- Introduces **age-appropriate** real-world knowledge about farming tools.
- Uses simple language accessible to children ages 3-6.
- Is **contextually relevant** to the story.

**Areas of Disagreement**

- The primary divergence is regarding the question's format and educational value:
    - **Emma Watson** (score: 5): Believes the simplicity is a strength, arguing that shorter, direct "what" questions are most effective for the 3–6 age range during interactive reading.
    - **Michael Chen** and **Sarah Thompson** (both score: 4): Identify opportunities for improvement in:
        * Modeling proper sentence structure and capitalization.
        * Using a more conversational framing to enhance **engagement**.
        * Adding follow-up questions to encourage deeper thinking.

**Overall Assessment** With an average score of 4.33, stakeholders generally view this question-answer pair positively while acknowledging room for improvement. The content appropriately bridges story elements with real-world knowledge in language accessible to young children. However, the educational value could be enhanced through better modeling of proper language structure and incorporating elements that stimulate deeper thinking or conversational engagement. The question succeeds in its basic educational function but misses opportunities to support more holistic language development that early childhood educators value.

A.6.2 EXAMPLE OF EVALUATION OUTPUTS FOR MSLR-COCHRANE

**Target Summary**: Evidence available to date does not allow any firm conclusions to be drawn concerning the effectiveness of ambulatory domicilary oxygen therapy in patients with COPD. Further studies are required in order to understand the role of ambulatory oxygen in the management of patients with COPD on long-term oxygen therapy. These studies should separate patients who desaturate from those who do not desaturate.

**Generated Summary**: There is some evidence that oxygen therapy may improve exercise performance in COPD. However, the evidence is based on a small number of small studies with variable quality of life outcomes. The results of this review are limited by the small number and small number available studies. Further research is needed to determine whether oxygen therapy is beneficial for exercise performance and whether it is safe and effective in COPDs.

**Averaged Human Score**: 0.50 (on 0-1)
**G-Eval** Score: 3.08 (1.00 after normalized to 0-1)
**ChatEval** Score: 4.00 (0.75 after normalized to 0-1)

**General Public**: Evaluation evidence: Assistant 1 provided a clear and concise summary, highlighting the current state of research and the need for further studies. However, Assistant 2 did not provide any summary, making it impossible to evaluate its performance.
The score of Assistant one: 4
The score of Assistant two: 0

**Critic**: Evaluation evidence: Assistant 1 provided a clear and concise summary, highlighting the current state of research and the need for further studies. However, it could have been more specific about the types of studies needed, as mentioned in the target summary. Assistant 2 did not provide any summary, making it impossible to evaluate its performance.
The score of Assistant one: 4
The score of Assistant two: 0

**MAJ-EVAL's Medical Researchers Group Score**: 3 (0.5 after normalized to 0-1)

**Overview** The evaluated summary is broadly accurate and grammatically sound, effectively communicating the uncertainty and limitations of current evidence regarding oxygen therapy in COPD.

All stakeholders agree that it appropriately conveys the need for further research and maintains a reasonable tone for a general overview.

**Areas of Consensus**

- **General Strengths**:
    - The summary is fluent.
    - It captures the uncertainty and limitations of current evidence.
- **Key Shortcomings**:
    - Lacks critical **clinical specificity**, particularly the **stratification of COPD patients** into desaturators and non-desaturators—considered essential for precision medicine, clinical decision-making, and evidence-based learning.
    - Omits **key terminology**, such as **"ambulatory domiciliary oxygen therapy"**, which is crucial for aligning with clinical guidelines and research frameworks.
    - **Conflation of outcomes**: The summary does not clearly distinguish between effectiveness outcomes (e.g., exercise performance) and safety outcomes, undermining clarity for clinical interpretation and learner education.

**Notable Nuance** Dr. Lukas Wagner emphasized an additional concern specific to **AI-based clinical decision support systems**—the **repetitive phrasing** (e.g., "small number") reduces clarity and could impair accurate processing by AI tools. This issue, while not highlighted by other stakeholders, is critical in contexts where evidence summaries are used to train or validate AI systems.

**Overall Interpretation** While the summary communicates the general uncertainty and limitations of current evidence effectively, it falls short of the precision and specificity required for clinical application, medical education, and AI-driven decision support. The consensus is that it is a **partially adequate** synthesis, but missing key elements significantly reduce its utility in practice and research.

**Average Score: 3.0** This score reflects a balanced view that the content is usable for general understanding but requires substantial refinement to meet the standards of clinical precision, educational utility, and technological integration.

### A.7 INTER-RATER RELIABILITY

We calculate the inter-rater reliability using Krippendorff's Alpha (K-Alpha) Krippendorff (2011) within each generated stakeholder group. The results are shown in Table 8.

### A.8 DETAILED WORKFLOW OF MAJ-EVAL'S PERSONA CREATION

Table 9 presents an example of MAJ-EVAL's persona creation process, including document selection, perspective extraction, and persona construction.

### A.9 EXAMPLES OF STAKEHOLDER PERSONA CREATION

For the Stakeholder Persona Creation phase, we show the extracted dimensions of the Parents' group (see Table 10), and the extracted dimensions of the Clinicians group (see Table 11), using Qwen-3-235B as the underlying model. Additionally, Tables 6 and 7 each illustrate one created personas for each stakeholder group on `StorySparkQA` and MSLR-COCHRANE, respectively.

### A.10 PROMPTS FOR BASELINE METHODS

#### A.10.1 PROMPTS FOR G-EVAL

Your task is to rate the `[evaluation content]` based on the **evaluation criteria** and the **task description**, following the specified **evaluation steps**.

Please make sure to read and understand these instructions carefully. Keep this document open while reviewing and refer to it as needed.

| Models | Stakeholder Group | Initial Eval | Final Eval |
|---|---|---|---|
| **StorySparkQA** | | | |
| Qwen3-235B | AI Developers | 0.46 | 0.25 |
| | Children | 0.52 | 0.39 |
| | Early Childhood Educators | 0.40 | 0.26 |
| | Language Researchers | 0.48 | 0.27 |
| | Parents | 0.40 | 0.29 |
| Claude-3.7-Sonnet | AI Developers | 0.39 | 0.70 |
| | Children | 0.34 | 0.52 |
| | Educational Experts | 0.56 | 0.67 |
| | Parents | 0.22 | 0.33 |
| | Teachers | 0.43 | 0.59 |
| **MSLR-COCHRANE** | | | |
| Qwen3-235B | Clinical Librarians | 0.55 | 0.50 |
| | Clinicians | 0.59 | 0.56 |
| | Evidence Synthesis Researchers | 0.54 | 0.54 |
| | Medical Researchers | 0.52 | 0.44 |
| Claude-3.7-Sonnet | Clinicians | 0.60 | 0.71 |
| | Emergency Physicians | 0.65 | 0.73 |
| | Interdisciplinary Clinicians | 0.76 | 0.83 |
| | Medical Review Experts | 0.68 | 0.81 |
| | Medical Researchers | 0.61 | 0.86 |
| | Public Health Consumers | 0.66 | 0.86 |

Table 8: Krippendorff's Alpha (K-Alpha) scores (based on inter-rater agreement) for each stakeholder group on the StorySparkQA and MSLR-COCHRANE datasets.

**Task Description:**
[task description]

**Evaluation Criteria:**
**Overall Quality**: The overall quality of the [evaluation content] should reflect all of the following dimensions: [Evaluation dimensions adapted from the original paper].

**Evaluation Steps:**

1. Read the source text carefully.

2. Read the AI-generated [evaluation content] and compare it to the [source text].

3. Assign a score for the overall quality (and other dimensions) of the AI-generated content using a 5-point Likert scale:
   - 1 – Strongly Disagree
   - 2 – Disagree
   - 3 – Neither Agree nor Disagree
   - 4 – Agree
   - 5 – Strongly Agree

**Source Text:**
[source text]

**AI-Generated Content:**
[evaluation content]

**Evaluation Form (Output Scores ONLY):**
Overall Quality: 1 / 2 / 3 / 4 / 5

| Stage | Step | Output |
|---|---|---|
| **Document Selection** | Search document keywords | *Example:* "Children reading with conversational agent qualitative interview", "Children interactive story reading CSCW" |
| | Documents found | *Example:* Sun et al. (2024), Chen et al. (2025b), Xu et al. (2021) |
| **Evaluative Dimension Extraction** | Stakeholder groups identified | Parents of preschool children, preschool children, educators, AI researchers |
| | Dimensions identified | *Example:* Parents expect AI-generated questions to be tailored to a child's cognitive level and psychological age, rather than being overly serious or professional. |
| | Evidence for each dimension | *Example:* "Our participants indicated that when they used tools such as C5 and C6 in Table 2 to answer children's story-related questions, the generated answers were often serious and professional, not specifically tailored to children's cognitive level and psychological age, which caused it difficult for children to understand." |
| **Persona Construction** | Persona constructed | *Example:* A parent who expects AI-generated questions to match his preschool daughter's comprehension level. He draws on his restaurant experience to explain complex ideas in child-friendly ways. Examples of full persona details (name, demographics, specialty, psychological traits, social relationships) are included in Table 6. |

Table 9: Overview of MAJ-EVAL's Persona Creation Workflow. Detailed examples of identified perspectives and created personas are presented in Appendix A.9.

### A.10.2    PROMPTS FOR CHATEVAL

For ChatEval, we used the original agent persona settings, with modified system prompt:

*[source text]*
...
*[The Start of Assistant 1's `[evaluation content]`]*
...
*[The End of Assistant 1's `[evaluation content]`]*
*[The Start of Assistant 2's `[evaluation content]`]*
...
*[The End of Assistant 2's `[evaluation content]`]*
*[System]*
We would like to request your feedback on the performance of the two AI assistants' generated `[evaluation content]` in response to the `[source text]` displayed above. Please focus your response on the utility of the QA pairs for the following task: `[task description]`. Assign an overall score for each assistant's QA pairs on a five-point Likert scale, with the following standards: 1 - Strongly Disagree; 2 - Disagree; 3 - Neither agree nor disagree; 4 - Agree; 5 - Strongly Agree

### A.11    TASK DESCRIPTIONS

**Question-Answer Generation in Children's Interactive Story-Reading** You need to evaluate the quality of AI-generated question-answer pairs from the storybook content. These AI-generated question-answer pairs are designed for the interactive storybook reading activity between parents and children aged 3 to 6, and should be grammatically correct and fluent in English. Parents expect to ask questions that are grounded in the storybook content, but introduce real-world common knowledge beyond the story content.

**Parent Characteristics:** Parents are primary caregivers from diverse educational and professional backgrounds who engage in interactive storybook reading activities with their children aged 3 to 6. These activities involve conversational exchanges designed to promote language development, with varying levels of skill, time, and motivation for interactive reading.

| Perspective | Evidence |
|---|---|
| Parents expect question-answer pairs grounded in story content but expanded to introduce real-world common knowledge to stimulate children's language and cognitive development. | Evaluation task defines this expectation explicitly in AI tool design. |
| Parents may not always engage in conversation-rich storybook reading due to skill, time, or inclination limitations. | Parents may not always pause the story, ask questions, and comment on their children's response, either assuming the child can learn well by listening or lacking skills or time for interactive opportunities. |
| Parents view AI tools as potential supports for interactive reading, provided they generate age-appropriate, grammatically correct English pairs that reduce parental burden while maintaining educational value. | Studies show parents expect AI to enhance storytelling through interactive elements that inspire connection and provide appropriate resources. |
| Parents value generation of contextually engaging, personalized question-answer pairs that align with children's cognitive stages and sustain meaningful educational interaction. | Current AI tools struggle with personalized, adaptive interaction that addresses children's developmental needs. |
| Parents expect questions to stimulate creativity, critical thinking, and curiosity rather than require rote factual recall, necessitating responses that adapt to children's exploratory thinking. | Parents criticize AI tools for being rigid and limiting active thinking development. |
| Parents emphasize the importance of maintaining grammatical correctness and speech-level appropriateness in AI-generated content to model proper language habits. | Parents identify overly complex vocabulary and adult-perspective questions as comprehension barriers. |

Table 10: Parental perspectives and supporting evidence extracted by MAJ-EVAL.

**Clinician Characteristics:** Medical professionals involved in patient care across various specialties who use biomedical literature to validate clinical suggestions and decisions. They require concise, yet detailed summaries that enable actionable decisions and consider patient-specific characteristics.

| Perspective | Evidence |
|---|---|
| Prioritize evidence from biomedical literature that aligns with the patient's specific characteristics (e.g., demographics, comorbidities) when validating AI suggestions. | Clinicians emphasized the importance of applicability of evidence to the patient's situation, such as matching demographics, comorbidities, and genetic factors (e.g., P8's case involving Liddle syndrome and family history). |
| Value comprehensiveness and reproducibility of evidence, ensuring both supporting and opposing studies are presented to avoid bias. | Clinicians curated comprehensive evidence lists, including all supporting and opposing studies and shared reproducible search links to validate AI suggestions. |
| Use the PICO framework (Population, Intervention, Comparator, Outcome) to assess the relevance of literature evidence for specific patient scenarios and demand specificity in reporting those elements. | Clinicians applied PICO to match patient populations with literature evidence, particularly in complex cases like rare diseases. Feedback emphasized the need for specificity in reporting, especially around PICO elements. |
| Prefer concise summaries of evidence that provide sufficient detail to ensure actionable decision-making, especially in time-constrained settings and request alerts for critical disagreements between AI suggestions and literature evidence. | Emergency care clinicians requested alerts for critical disagreements between AI suggestions and literature evidence, balancing conciseness with clinical urgency. |

Table 11: Clinician perspectives and supporting evidence extracted by MAJ-EVAL.

**Multi-document Summarization in Medicine** You need to evaluate the quality of AI-generated short biomedical summaries that integrate findings from multiple literature reviews. These grammatically correct and fluent summaries are designed to help medical professionals efficiently capture the key findings across studies and provide a coherent overview of relevant medical questions and outcomes (just like the target summary).

## A.12 PROMPTS OF MAJ-EVAL

To utilize LLMs' strong reasoning and generation capability as well as control LLMs' outputs as much as possible to meet the needs of diverse evaluation task requirements, we carefully design our prompts. Table 12, 13, 14, 15, 16, and 17 list all the prompts we used for MAJ-EVAL.

### A.12.1 PROMPTS FOR STAKEHOLDER PERSONA CREATION

Table 12 and 13 list the prompts we used for the creation of multi-stakeholder personas.

### A.12.2 PROMPTS FOR DEBATING

Table 14, 15, 16, and 17 list the prompts we used for agents' in-group debate.

---

**Prompt for Identifying Stakeholder Perspectives**

---

You need to identify or construct a diverse and comprehensive set of stakeholders, their characteristics, and their perspectives or opinions for the following evaluation task:
**{task_description}***

**Guidelines**
- For this given paper, read one paragraph at a time. Ignore the related work section and literature list.
Step 1 - Identify *ALL* mentioned name entities, excluding the authors and their institutions, as well as non-human entities.
Step 2 - For each name entity (i.e., stakeholder) you identified, generate the descriptive characteristics for this stakeholder. Then extract their perspectives or opinions that are **relevant to the aforementioned evaluation task**. Each entry should be directly derived from the texts with supporting evidence.

**Important Reminders**
- If in the provided paper, no relevant information is mentioned about the evaluation task, output nothing.
- In generation, prioritize capturing a wide range of stakeholders and their perspectives, including those that might emerge from different roles, backgrounds, and needs.
- The stakeholder's perspectives or opinions should be relevant to the aforementioned evaluation task.
- Each final generated stakeholder entry should clearly include:
    1. The stakeholder name (e.g., role or representative group),
    2. The stakeholder's characteristics,
    3. The stakeholder's perspectives or opinions regarding the aforementioned evaluation task,
    4. The supporting evidence from the provided papers.

**Output Format**
- If the provided paper contains relevant information about the evaluation task, present the output as a structured JSON dict, with each item formatted as an object containing the following fields: {

```
    "stakeholder name": {
        "characteristics": "use one sentence to describe the
stakeholder's characteristics",
        "perspectives": [
            {
                "perspective": "use one sentence to describe
the stakeholder's perspectives or opinions",
                "evidence": "supporting evidence from the
provided paper"
            },
            { ... }
        ]
    },
    "stakeholder name": { ... }
}
```

- If no relevant information is found: `[]`

Table 12: Prompt for identifying stakeholder perspectives based on provided paper literature and the evaluation task description.

---

**Prompt for Constructing Stakeholder Personas**

---

You need to create stakeholder personas for the following evaluation task:
**{Task Description}***

**Guidelines**
- For the provided stakeholder perspective list, process one stakeholder at a time.
- For each mentioned perspective of the stakeholder, generate a distinct persona that embodies the corresponding perspective.
- Following the steps below, each generated persona must include these attributes:
1. Generate the persona's `demographic information` based on name, age, education, career, personality traits, hobbies, etc.
2. Rephrase the stakeholder `perspective` to match the persona.
3. Generate a `specialty` aligned with the persona's profile and relevant to the evaluation task.
4. Generate `psychological traits` describing personality, emotions, and cognitive tendencies.
5. Generate the persona's `social relationships` that reflect connections within the stakeholder types.

**Important Reminders**
- Personas should be diverse, realistic, and grounded in the stakeholder profile.
- Each distinct perspective must map to a unique persona.

**Stakeholder Perspective List**
{Identified Stakeholder Perspectives}

**Output Format** (as JSON structure):
```
{
 "Stakeholder Name":  [
  {
 "Name":  "Full name of the persona",
 "Demographic Information":  "One to two sentences describing
the persona's demographic profile.",
 "Perspective":  "One to two sentences outlining the persona's
perspective.",
 "Specialty":  "One to two sentences describing the persona's
skill or expertise.",
 "Psychological Traits":  "One to two sentences describing
personality, emotions, etc.",
 "Social Relationships":  "One to two sentences describing
interactions with other stakeholders."
  },
  { ...  }
  ],
  "Another Stakeholder Name":  [ ...  ]
}
```

Table 13: Prompt for generating stakeholder personas grounded in identified perspectives.

---

**Prompt for Instantiating Stakeholder Agent**

---

YOU ARE {agent_name}. Your demographic information is: {}.
Your perspective is: {}.
Your specialty is: {}.
Your psychological traits include {}.
Socially, these are your relationships: {}.

Using your perspective and/or specialty, now you are evaluating the quality and appropriateness
of AI-generated candidate {evaluation content} for the following task:
**{Task Description}***

The content to be evaluated is: {}
The related context for the evaluation content is: {}
You should use this format for your evaluation: {}

**Follow the steps below:**
1. In phase 1 of the evaluation, you need to generate your initial evaluation result.
2. In phase 2 of the evaluation, there are other stakeholders with different specialties
who are also doing the same evaluation task, and you will participate in a debate.
During debate, you will express your opinions and listen to others' perspectives to decide
whether you should change your evaluation decision.

When others express their feedback, reflect on their input from your own perspective.
Consider whether their viewpoints reveal aspects you may have overlooked.
If others comment on your evaluation, you should reflect on your evaluation and decide
whether to accept others' comments. However, you do not need to agree with others.
You must base your evaluation on your own perspective and/or specialty.

When it's your turn to speak, you may:
- Offer comments or critiques on previous feedback if you find any issues or meaningful
contrasts.
- If you find all prior evaluations reasonable and have no further comments, respond with "NO
MORE COMMENTS" and provide your final evaluation in the aforementioned format.

**Important Reminder:**
Your feedback and score must remain grounded in your own perspective and/or area of expertise.
Do not generate evaluations that duplicate or closely mirror those of other agents.

---

Table 14: Prompt for stakeholder-grounded agent evaluation with structured debate and reflection.

---

**Prompt for Debating Phase 1: Independent Initial Feedback**

---

You are now in **Phase 1** of the evaluation process. You need to provide your initial feedback and score of the content based on your perspective and/or specialty.

The content to be evaluated: {}
The related context for the evaluation content: {}
Response format: {}

**Instructions:**
- Your evaluation should reflect your own unique perspective and area of expertise.
- Focus on assessing the quality and appropriateness of the content for the given evaluation scenario.
- Your response should use the exact format provided above.

**Important Reminder:**
Do not replicate evaluations from others. Stay grounded in your own perspective.

---

Table 15: Prompt for Phase 1 in the multi-agent in-group debate evaluation: agents provide their initial judgment.

---

**Prompt for Debating Phase 2: Free Debate**

---

**1. Debate Start:**
You are now entering **Phase 2** of the evaluation process, where you need to participate in a debate process with other stakeholders like you.

Here are the initial evaluations from all stakeholders: {phase 1 evaluations}
Your task is to evaluate these initial assessments based on your perspective and/or specialty. You should also reflect on the feedback from other stakeholders and decide whether to agree, disagree, or add nuances to the discussion based on your perspective and/or specialty.

**2. During Debate:**
Now, it's your turn to speak. Based on all previous feedback from the debates and your reflection, you can decide whether to agree, disagree, or add nuances to the discussion based on your perspective and/or specialty.

If you have no more points to discuss, respond with "NO MORE COMMENTS" followed by your final evaluation in this format: {response format}

**Important Reminder:**
Your feedback and score should be based on your perspective and/or specialty. Avoid generating evaluations that duplicate or closely mirror those of other agents.

---

Table 16: Prompt for Phase 2 in the multi-agent in-group debate evaluation: free debate.

**Prompt for Final Aggregation of Multi-Stakeholder Evaluations**

You are an impartial evaluation aggregator. Your task is to review the evaluations from multiple stakeholders and provide a comprehensive summary that fairly represents all perspectives.

Your summary should include key areas of agreement and disagreement, and an overall assessment that reflects the range of perspectives.

You are given all final evaluations in {`aggregated_content`} and their average score in {`average_score`}.

Format your response as a JSON object with the following structure:
```
{
    'Feedback':  'A clear, concise synthesis of stakeholder
feedback, highlighting consensus, divergence, and an overall
interpretation.',
    'Average Score':  x
}
```

Table 17: Prompt for Phase 3 in the multi-agent in-group debate evaluation: aggregating final evaluations across stakeholder agents into a unified summary.

