# OpenReview forum: "Multi-Agent-as-Judge: Aligning LLM-Agent-Based Automated Evaluation with Multi-Dimensional Human Evaluation"
_ICLR.cc/2026/Conference — Submitted to ICLR 2026_

### Official Review · Reviewer_uxh7 · 2025-10-31

**Soundness:** 3
**Presentation:** 3
**Contribution:** 3
**Rating:** 4
**Confidence:** 3

**Summary:**

This paper introduces MAJ-EVAL, a multi-agent LLM-based evaluation framework designed to simulate real-world multi-stakeholder assessment of NLG systems. The method automatically extracts stakeholder perspectives from domain-specific documents, constructs detailed evaluator personas, and orchestrates in-group debates among these agents to produce multi-dimensional evaluation scores. Experiments on two domains—children’s storybook QAG and medical literature summarization—demonstrate that MAJ-EVAL achieves significantly higher alignment with human expert ratings than automated metrics (ROUGE-L, BERTScore), single-LLM evaluation (G-Eval), and prior multi-agent methods (ChatEval). The ablation results confirm that both persona construction and in-group debate contribute to improved human correlation. The framework generalizes across domains and offers a systematic way to ground LLM evaluators in real human roles.

**Strengths:**

- The idea of grounding evaluator personas in real stakeholder perspectives extracted from research documents is original and methodologically well-motivated.
- Results across two distinct domains (education and medicine) convincingly show the framework’s generalizability.
- The paper carefully isolates the effects of persona creation and debate, demonstrating clear improvements in human correlation.
- The work moves beyond surface metrics, emphasizing authentic alignment with how people evaluate complex, domain-specific outputs.

**Weaknesses:**

- MAJ-EVAL involves multiple LLM agents and debate iterations, which may be expensive and time-consuming in practice.
- Only two domains were tested; more varied settings (e.g., legal, creative writing) would strengthen claims of generalizability. Both tasks (StorySparkQA and MSLR-Cochrane) are well-defined and structured; it’s unclear whether the framework can handle noisy, open-ended real-world outputs.
- The framework assumes access to well-structured, domain-relevant papers for persona extraction—an assumption that may not hold in under-documented domains.
- Correlation with human ratings is the only quantitative metric reported; qualitative validation of the debate process itself is limited.
- Since the objective is maximizing correlation with human scores, it’s unclear whether MAJ-EVAL always captures better reasoning or merely reproduces human biases.
- The paper doesn’t deeply analyze how or why agents converge during debates—leaving open whether the final scores reflect genuine consensus or averaging artifacts.

**Questions:**

- How scalable is MAJ-EVAL when applied to large datasets or high-throughput evaluation pipelines?

- Can the framework operate effectively when domain documents are scarce or noisy, for instance, in emerging or low-resource fields?

- How sensitive are the results to the underlying base model (e.g., Claude vs. Qwen), and do persona representations transfer across models?

- Did the authors analyze whether in-group debates ever lead to groupthink effects, where agents converge prematurely rather than explore diverse viewpoints?

---

> ### Author Response · Authors · 2025-12-04
> **Response to Your Feedback (Part 1 of 2)**
>
> Thank you for your insightful comments on our work! We appreciate the opportunity to address your concerns regarding our work.
>
> **1. Potential issue with under-documented domains**
>
> We acknowledge that document availability varies across domains. Our current work focuses on domains where stakeholder documentation is already common (e.g., education guidelines, medical reporting standards, institutional protocols), as these materials directly inform how human experts evaluate system outputs. However, we would like to clarify that MAJ-Eval can leverage any available stakeholder source, rather than relying solely on formal academic papers. Even in less-documented domains, stakeholder perspectives are often reflected through alternative sources, such as community forums and expert interviews. We will discuss potential future work to explore MAJ-Eval’s applicability across under-documented domains.
>
> **2. Do agents converge during debates?**
>
> We understand your concern. To clarify, MAJ-Eval does not aim to force stakeholder agents to converge to a single “correct” answer, because real-world stakeholders often maintain diverse and even conflicting priorities. Instead, the debate phase allows stakeholders within the same group to fully discuss, challenge, and refine their judgments based on what they care about most. For example, in the children’s QAG task, educational experts naturally focus more on developmental appropriateness and instructional value. During debate, edu expert agents actively elaborate on these concerns, leading to higher correlations to the “Educational Appropriateness” dimension. Therefore, the debate mechanism improves human alignment not by enforcing convergence, but by exchanging and refining stakeholder-specific evaluation perspectives.
>
> **3. Missing qualitative validation of the debate**
>
> We have included qualitative analysis in Appendix A.6. Very briefly, the qualitative analysis shows that the debate phase enables agents to surface domain-specific dimensions beyond current human evaluation dimensions.
>
> **4. Additional domains and underlying models**
>
> We would like to clarify that the primary contribution of MAJ-Eval lies in its ability to automatically generalize across specialized domains. The two specialized domains we chose (i.e., children’s education and medicine) represent popular research areas in NLP with sufficient available datasets. Also, these two domains totally differ in stakeholder perspectives, evaluation goals, and criteria. Therefore, MAJ-Eval’s consistent stronger performance on these two domains demonstrate its generalizability. We acknowledge that the availability of human-annotated data in other specialized domains is still limited, and we have noted this limitation in Section 8. In future work, we plan to incorporate additional tasks to further illustrate the generalizability of MAJ-Eval.
>
> In addition, we fully agree that testing more backbone models would further strengthen the backbone-agnostic claim. Our goal in this work, however, was not to exhaustively benchmark all LLMs, but to show that MAJ-Eval remains stable across distinct backbone types. Therefore, we selected two strong yet architecturally different models (Qwen and Claude) within our computational budget, as they represent the most common deployment settings in practice (i.e., a high-performing open-source model and a commercial model). Benchmarking additional models would incur considerably higher inference costs, and thus was not prioritized in this initial study. As future work, we plan to expand the backbone comparison and include lightweight and emerging models to further broaden robustness evidence.
>
> **5. Clarification on cost**
>
> We acknowledge that MAJ-Eval is not inexpensive, but it is still far more economical and convenient than hiring domain experts (see our cost analysis in Appendix A.5). Based on our records, the total token usage per task is roughly 141,329 tokens. At Claude 3.7 Sonnet’s pricing of `$3` per million tokens, the cost is roughly `$0.42`, which is significantly lower than recruiting human experts for evaluation. Since MAJ-Eval is designed to supplement existing evaluation methods with a more human-aligned and efficient alternative, we believe its time and monetary cost are reasonable and significantly more scalable than expert-based evaluation.

---

> ### Author Response · Authors · 2025-12-04
> **Response to Your Feedback (Part 2 of 2)**
>
> **6. Would correlation with human judgements introduce bias?**
>
> We agree that traditional human evaluation in NLP can introduce bias because the ratings may reflect personal preferences and the models could have inherent bias. However, we try to reduce human bias by grounding evaluative dimensions from qualitative research publications, as HCI and CSCW research commonly understand stakeholder perspectives through qualitative interviews, where users explain why they value certain system behaviors and what constraints they face in practice. This qualitative method could help surface shared, user-centered evaluation criteria and reduce reliance on isolated personal opinions. In future work, we will further investigate the fairness of human evaluative dimensions.
>
> In addition, we added an ablation using LLM-generated personas to engage in an in-group debate, in order to validate MAJ-Eval's document-grounded persona construction process.  The results are shown in the following table:
>
> | Method  | Overall Quality | Grammar Correctness | Answer Relevancy | Contextual Consistency | Educational Appropriateness |
> | ----------------------------------------- | --------------- | ------------------- | ---------------- | ---------------------- | --------------------------- |
> | LLM-Generated Persona (Claude-3.7-Sonnet) | 0.37  | **0.17**               | 0.39             | 0.20            | 0.32        |
> | MAJ-Eval (Claude-3.7-Sonnet)              | **0.47**            | 0.14   | **0.45**             | **0.33**             | **0.40**                  |
> | LLM-Generated Persona (Qwen-3-235B)       | 0.34            | 0.15                | 0.42             | 0.15         | 0.20                        |
> | MAJ-Eval (Qwen-3-235B)                    | **0.43**        | **0.18**            | **0.43**             | **0.27**      | **0.33**               |
>
> | Method                                    | Overall Quality | Fluency | PIO Consistency | Effect Direction | Evidence Strength |
> | ----------------------------------------- | --------------- | ------- | --------------- | ---------------- | ----------------- |
> | LLM-Generated Persona (Claude-3.7-Sonnet) | 0.32         | **0.14**    | 0.16            | 0.26             | 0.20              |
> | MAJ-Eval (Claude-3.7-Sonnet)              | **0.40**         | 0.10    | **0.21**            | **0.34**             | **0.28**              |
> | LLM-Generated Persona (Qwen-3-235B)       | 0.27            | 0.11    | 0.14            | 0.14             | **0.24**              |
> | MAJ-Eval (Qwen-3-235B)                    | **0.39**            | **0.34**    | **0.21**            | **0.26**             | 0.18              |
>
> As shown in the table, **MAJ-Eval consistently outperforms the baseline that relies on LLM-generated personas**, both in overall quality and in domain-specific dimensions such as Educational Appropriateness and Effect Direction. Qualitatively, we find that LLM-generated personas often fail to construct a comprehensive set of stakeholder groups. For example, for the medical-summarization task, Qwen generated only three stakeholder groups: Clinicians, Biomedical Researchers, and Healthcare Administrators/Policymakers. However, the actual stakeholders are more diverse, including Clinical Librarians, Public Health Consumers, etc. We will incorporate these additional baseline results and our expanded analysis into the revised version.

---

### Official Review · Reviewer_7tKk · 2025-11-01

**Soundness:** 2
**Presentation:** 3
**Contribution:** 2
**Rating:** 4
**Confidence:** 5

**Summary:**

The paper proposes MAJ-EVAL, a framework for large-scale text evaluation using multi-agent debates among simulated stakeholder personas. It constructs evaluator personas derived from literature, lets them discuss through structured debates, and aggregates their final opinions. Experiments on creative writing and medical summarization benchmarks show that MAJ-EVAL’s aggregated judgments align more closely with human evaluations than single-model or self-consistency baselines.

**Strengths:**

The paper makes a timely contribution to LLM-based evaluation. I like that the authors automated agent generation step, which will improve replicability and objectivity. The results show improved reliability, making the approach conceptually novel and practically relevant for human-aligned evaluation design.

**Weaknesses:**

Comment 1. The authors might want to check whether their persona extraction is valid. Several additional experiments could mitigate this concern:
(1) The authors can randomly perturb the input corpus and track downstream ρ/τ changes against human ratings.
(2) Although costly, the authors might consider recruiting several domain experts and asking them to verify the extracted dimensions.
Additionally, domain experts can rate persona faithfulness and coverage. The authors would then be able to report the precision or recall of model dimensions relative to expert lists.

Comment 2. Several recent papers (e.g., see Choi et al., 2025) show that multi-agent debate may lead to only incremental gains. It is important to understand where the authors’ performance gains come from—do they stem from multi-agent debates or simple aggregation?
The authors can report (i) no-debate means of independent agents, (ii) “read-others-but-no-discussion,” and (iii) self-consistency ensembling (multiple samples per agent, no debate).
The authors also need to report the token usage of each method so that users can assess the performance–cost trade-off (in addition to the discussion in A5).
Ref: Choi, Hyeong Kyu, Xiaojin Zhu, and Yixuan Li. "Debate or Vote: Which Yields Better Decisions in Multi-Agent Large Language Models?" arXiv preprint arXiv:2508.17536 (2025).

Comment 3. Another interesting ablation could involve using the expert dimensions extracted from the first domain on the other domain (and vice versa). This would directly show the gains achieved by automating the agent (and persona) generation process.

Comment 4. I am not sure why the authors include demographic characteristics in their personas. This may inadvertently drive scoring beyond evaluative dimensions, especially if certain demographics correlate with stricter or more lenient judgments.
Although unlikely, it is worth checking (at least on a subsample) (i) agents with no demographic characteristics and (ii) randomizing demographic characteristics only.

Comment 5. There are several missing key citations, including:
Kumar, Sandeep, Abhijit A. Nargund, and Vivek Sridhar. "CourtEval: A Courtroom-Based Multi-Agent Evaluation Framework." Findings of the Association for Computational Linguistics: ACL 2025 (2025).
Kim, Alex, Keonwoo Kim, and Sangwon Yoon. "DEBATE: Devil's Advocate-Based Assessment and Text Evaluation." arXiv preprint arXiv:2405.09935 (2024).
Du, Yilun, et al. "Improving Factuality and Reasoning in Language Models through Multi-Agent Debate." Forty-first International Conference on Machine Learning (2023).
I recommend that the authors conduct a comprehensive literature review and include more relevant, recent citations.

Comment 6. The citation format should be changed. In-line citations should appear as (AAA, 2025), not AAA (2025).

**Questions:**

Please see above (weaknesses).

---

> ### Author Response · Authors · 2025-12-04
> **Response to Your Feedback (Part 1 of 2)**
>
> We sincerely value your time reviewing our work, and we would like to provide some further clarifications:
>
> **1. Verification on persona extraction**
>
> We totally agree with you that the extracted persona should be consistent with the source documents. To clarify, in our implementation, the persona creation is divided into two steps: (1) Evaluative Dimension Extraction and (2) Persona Construction. During dimension extraction, we explicitly prompt the model to provide both the dimension and the corresponding source text as evidence. We then compute textual similarity between the evidence and the document segments to verify that each dimension is directly grounded in the source literature. Because persona construction is based strictly on these validated dimensions, the resulting personas remain grounded in the original documents.
>
> **2. Understanding of where the performance gains come from**
>
> We have reported the no-debate baseline in Figures 4 and 5, denoted as the initial evaluation results. Comparing pre-debate and post-debate outcomes reveals that MAJ-Eval’s high correlation with human judgments largely comes from its stakeholder-grounded evaluative dimensions extracted during persona creation. However, our qualitative analysis (see Appendix A.6) shows that the debate mechanism surfaces additional dimensions that are not captured by existing human evaluation dimensions but are highly relevant to stakeholder needs. As such evaluation dimensions are well recognized in stakeholder-centered evaluation in CSCW and HCI research [1], we believe the debate mechanism is beneficial to facilitate richer, stakeholder-centered reasoning aligned with real-world evaluation practices.
>
> **3. Ablation study with cross-domain metrics**
>
> Thank you for the suggestion. We would like to clarify that, since different domain-specific scenarios are often unique, the stakeholders’ evaluative dimensions typically vary across tasks. While MAJ-Eval’s method is generalizable, the evaluation metrics are specific to each domain. Applying dimensions from one domain to another would not meaningfully test our framework, but would instead create a domain mismatch (e.g., Educational Appropriateness is irrelevant to medical summarization, where Medical Evidence Strength is more essential).
>
> In response to your suggestion for an ablation study, we conducted an additional ablation using LLM-generated personas to engage in an in-group debate, in order to validate MAJ-Eval's persona construction process.  The results are shown in the following table:
>
> | Method  | Overall Quality | Grammar Correctness | Answer Relevancy | Contextual Consistency | Educational Appropriateness |
> | ----------------------------------------- | --------------- | ------------------- | ---------------- | ---------------------- | --------------------------- |
> | LLM-Generated Persona (Claude-3.7-Sonnet) | 0.37  | **0.17**               | 0.39             | 0.20            | 0.32        |
> | MAJ-Eval (Claude-3.7-Sonnet)              | **0.47**            | 0.14   | **0.45**             | **0.33**             | **0.40**                  |
> | LLM-Generated Persona (Qwen-3-235B)       | 0.34            | 0.15                | 0.42             | 0.15         | 0.20                        |
> | MAJ-Eval (Qwen-3-235B)                    | **0.43**        | **0.18**            | **0.43**             | **0.27**      | **0.33**               |
>
> | Method                                    | Overall Quality | Fluency | PIO Consistency | Effect Direction | Evidence Strength |
> | ----------------------------------------- | --------------- | ------- | --------------- | ---------------- | ----------------- |
> | LLM-Generated Persona (Claude-3.7-Sonnet) | 0.32         | **0.14**    | 0.16            | 0.26             | 0.20              |
> | MAJ-Eval (Claude-3.7-Sonnet)              | **0.40**         | 0.10    | **0.21**            | **0.34**             | **0.28**              |
> | LLM-Generated Persona (Qwen-3-235B)       | 0.27            | 0.11    | 0.14            | 0.14             | **0.24**              |
> | MAJ-Eval (Qwen-3-235B)                    | **0.39**            | **0.34**    | **0.21**            | **0.26**             | 0.18              |
>
> As shown in the table, **MAJ-Eval consistently outperforms the baseline that relies on LLM-generated personas**, both in overall quality and in domain-specific dimensions such as Educational Appropriateness and Effect Direction. Qualitatively, we find that LLM-generated personas often fail to construct a comprehensive set of stakeholder groups. For example, for the medical-summarization task, Qwen generated only three stakeholder groups: Clinicians, Biomedical Researchers, and Healthcare Administrators/Policymakers. However, the actual stakeholders are more diverse, including Clinical Librarians, Public Health Consumers, etc. We will incorporate these additional baseline results and our expanded analysis into the revised version.

---

> ### Author Response · Authors · 2025-12-04
> **Response to Your Feedback (Part 2 of 2)**
>
> **4. Inclusion of demographic information**
>
> Thank you for raising this concern. In MAJ-Eval, demographic information is primarily used as contextual cues that support realistic role-playing. This design choice is grounded in prior research showing that personas with background information enable agents to generate more contextually appropriate and situated simulations, rather than generic responses [2][3]. In addition, in the debating phase, each agent is explicitly instructed to ground their evaluation on their evaluative perspectives. As evidenced by our qualitative analysis and sample outputs (Appendix A.6), agents consistently anchor their judgments in these perspectives rather than in demographic information.
>
>
>
> [1] Berman, Glen et al. “A Scoping Study of Evaluation Practices for Responsible AI Tools: Steps Towards Effectiveness Evaluations.” Proceedings of the 2024 CHI Conference on Human Factors in Computing Systems (2024): n. pag.
>
> [2] Park, J. S., Zou, C. Q., Shaw, A., Hill, B. M., Cai, C., Morris, M. R., ... & Bernstein, M. S. (2024). Generative agent simulations of 1,000 people. arXiv preprint arXiv:2411.10109.
>
> [3] Jiang, H., Zhang, X., Cao, X., Breazeal, C., Roy, D., & Kabbara, J. (2024, June). PersonaLLM: Investigating the ability of large language models to express personality traits. In Findings of the association for computational linguistics: NAACL 2024 (pp. 3605-3627).

---

### Official Review · Reviewer_Ta4f · 2025-11-02

**Soundness:** 3
**Presentation:** 3
**Contribution:** 3
**Rating:** 4
**Confidence:** 3

**Summary:**

The paper proposes **MAJ‑EVAL**, a two‑stage framework for “multi‑agent‑as‑judge” evaluation that aims to align LLM‑based automatic judgments with multi‑stakeholder, multi‑dimensional human evaluations in real‑world tasks. Stage 1 automatically constructs stakeholder‑grounded personas by extracting evaluative dimensions (with evidence) from domain documents and then instantiating rich persona profiles (demographics, specialty, psychological traits, social relationships). Stage 2 runs in‑group, free‑form debates among personas and aggregates their judgments into per‑dimension scores. The approach is evaluated on two domains: children’s QAG and medical multi‑document summarization, reporting higher correlations with human ratings than traditional metrics (ROUGE/BERTScore), single‑judge and existing multi‑agent baselines.

**Strengths:**

1. **Stakeholder‑grounded automatic persona creation.** The two‑step procedure (dimension extraction → persona instantiation with rich attributes) is novel in the evaluation context and increases face validity of agents’ judgments relative to ad‑hoc, hand‑written personas.
2. **Consistent human alignment on multiple dimensions.** Across StorySparkQA and MSLR‑Cochrane, MAJ‑EVAL achieves stronger correlations with human ratings than ROUGE/BERTScore, single‑LLM judges, and a prior multi‑agent baseline, especially on domain‑specific dimensions (e.g., Educational Appropriateness; Evidence Strength).
3. **Debate mechanism that improves judgments.** The in‑group free debate with a moderator and final aggregation is well‑motivated and empirically increases average correlations for most stakeholder groups (pre‑ vs. post‑debate).
4. **Reproducibility‑oriented reporting.** Prompts, pipeline summaries, and ablations are documented; the paper includes an anonymized code link and describes token budgeting and setup to aid reproduction.

**Weaknesses:**

1. **Judge backbones are limited.** Experiments use **only two** judge LLMs (Qwen‑3‑235B and Claude‑3.7‑Sonnet). This leaves open whether the gains are robust across families/scales (e.g., Llama‑3.x, GPT‑4‑series, Mistral‑Large).
2. **Correlation differences lack uncertainty analysis.** Several comparisons rely on visual gaps in heatmaps/tables. Confidence intervals, statistical tests (e.g., Zou’s method for comparing dependent correlations), or bootstrap CIs would substantiate claims.
3. **Potential dependence on selected literature for persona construction.** The dimension‑extraction step may inherit biases/coverage gaps from the chosen documents. The paper would benefit from reporting selection protocols, diversity checks, and sensitivity to the literature pool.
4. **Ablations could be deeper.** While simple‑role vs. detailed persona and pre‑ vs. post‑debate are useful, the work lacks controlled studies on **persona count**, **group size**, **debate length/turn budget**, and **moderation strategies**.
5. **Cost and latency trade‑offs need fuller treatment.** The token costs reported are substantial; per‑instance cost and comparisons to cheaper single‑judge settings would help practitioners decide when MAJ‑EVAL is worth it.
6. **External validity still limited.** Only two domains/tasks are studied; transfer to other NLG settings (dialogue safety, instruct‑following, data‑to‑text) remains to be shown.

**Questions:**

1. **Backbone robustness:** Can you add results with at least one *open* and one *closed* additional judge (e.g., Llama‑3.1‑70B‑Instruct, GPT‑4o‑mini) to test backbone‑agnosticism? (Sec. 4.4 shows only Qwen/Claude.)
2. **Uncertainty reporting:** Please provide confidence intervals for key correlations and, where relevant, statistical tests for correlation differences (dependent correlations across methods on the same items).
3. **Literature pool & sensitivity:** How are domain documents collected for dimension extraction (inclusion/exclusion criteria, #papers, time ranges)? Any sensitivity analysis when random subsets of papers are used?
4. **Ablation on debate budget:** What is the effect of limiting debate turns, changing moderator heuristics, or varying agent counts per stakeholder group?
5. **Calibration & scale:** What exact rating scales are used internally by agents per dimension, and how are they normalized before aggregation (cf. 5‑point Likert in A.10 and 0–1 normalization in Table 5, p.19)?

---

> ### Author Response · Authors · 2025-12-04
> **Response to Your Feedback (Part 1 of 3)**
>
> We truly appreciate your insightful suggestions and encouragement to strengthen our work. We would like to take the opportunity to provide further clarifications and details regarding our work:
>
> **1. Clarification on document selection (inclusion/exclusion criteria, #papers, time ranges)**
>
> To clarify, MAJ-Eval’s document selection process is semi-automated, reproducible, and guided by explicit inclusion criteria: First, we identify qualitative studies cited by the dataset authors. Second, we perform a keyword-based search on Google Scholar, combining task-specific terms with “qualitative interview”, and restrict results to publications from the past three years. Based on pilot testing, we select two to three documents per task, which provides sufficient perspective diversity while maintaining coherent persona construction (example shown in Appendix Table 9).  We will revise Section 4.4 to clearly describe these inclusion criteria and the recommended document quantity.
>
> We also acknowledge that no literature pool can fully cover all possible stakeholder dimensions. Nevertheless, the dimensions extracted by MAJ-Eval not only align with but also supplement existing human evaluation dimensions. We view MAJ-Eval as a step toward more user-grounded evaluation rather than a complete solution. As future work, we plan to connect MAJ-Eval with broader knowledge resources, enable automated queries, and conduct broader sensitivity analyses to expand stakeholder coverage.
>
> **2. Additional experiment and analysis**
>
> Thank you for bringing the lack of uncertainty analysis to our attention. In response, we have added p-values and bootstrap confidence intervals (10,000 bootstrap resamples, 95% CI). The table below summarizes the uncertainty statistics for the evaluation results generated by Claude-3.7-Sonnet on the StorySparkQA dataset.
>
> | Stakeholder Group       | Grammar Correctness | Answer Relevancy    | Contextual Consistency | Educational Appropriateness | Overall Quality     |
> | ----------------------- | ------------------- | ------------------- | ---------------------- | --------------------------- | ------------------- |
> | **AI Developers**       | 0.14 [-0.16, 0.24]  | 0.43** [0.25, 0.60] | 0.31* [0.04, 0.50]     | 0.37** [0.12, 0.61]         | 0.42** [0.24, 0.64] |
> | **Children**            | 0.11 [-0.18, 0.28]  | 0.43** [0.27, 0.63] | 0.29** [0.03, 0.47]    | 0.35 [0.13, 0.60]           | 0.41** [0.25, 0.65] |
> | **Educational Experts** | 0.11 [-0.03, 0.32]  | 0.40** [0.22, 0.57] | 0.33** [0.14, 0.52]    | 0.36** [0.17, 0.59]         | 0.45** [0.31, 0.65] |
> | **Parents**             | 0.17 [-0.19, 0.27]  | 0.42** [0.23, 0.57] | 0.27** [0.07, 0.47]    | 0.35** [0.10, 0.60]         | 0.41** [0.23, 0.63] |
> | **Teachers**            | 0.12 [-0.10, 0.27]  | 0.40** [0.25, 0.60] | 0.36** [0.20, 0.53]    | 0.40** [0.23, 0.62]         | 0.47** [0.35, 0.69] |
> | **Overall**             | 0.11 [-0.12, 0.28]  | 0.45** [0.28, 0.63] | 0.33** [0.12, 0.52]    | 0.40** [0.19, 0.63]         | 0.47** [0.32, 0.69] |
>
> \* denotes p < .05 and ** denotes p < .01. Values in brackets indicate 95% confidence intervals.
>
> Across all stakeholder groups, correlations for Grammar Correctness are low and not statistically significant, while Answer Relevancy, Contextual Consistency, Educational Appropriateness, and Overall Quality all show moderate, statistically significant correlations, with 95% CIs that exclude zero. Notably, the strongest alignment appears in Overall Quality and Educational Appropriateness, especially for educational experts and teachers, indicating that **MAJ-Eval captures stakeholder-grounded, domain-specific judgments more reliably than surface-level metrics**.
>
> In addition, we fully agree that testing more backbone models would further strengthen the backbone-agnostic claim. Our goal in this work, however, was not to exhaustively benchmark all LLMs, but to show that MAJ-Eval remains stable across distinct backbone model types. Therefore, we selected two strong yet architecturally different models (Qwen and Claude) within our computational budget, as they represent the most common deployment settings in practice (i.e., a high-performing open-source model and a commercial model). Benchmarking additional large models such as Llama-3.1-70B-Instruct or GPT-4o-mini would incur considerably higher inference costs, and thus was not prioritized in this initial study. As future work, we plan to expand the backbone comparison and include lightweight and emerging models to further broaden robustness evidence.

---

> ### Author Response · Authors · 2025-12-04
> **Response to Your Feedback (Part 2 of 3)**
>
> **3. Additional ablation study on persona number, debate turns, etc.**
>
> We would like to clarify that we did not manually manipulate the number of personas because persona generation in MAJ-Eval is fully automated and directly determined by the content extracted from the provided documents. In pilot testing, we ensured that MAJ-Eval consistently identifies a comprehensive set of stakeholders from relevant literature. Artificially removing personas would contradict the framework’s design goal of document-grounded persona construction. For debate turns, we set the maximum at three turns. Our experiment logs show that stakeholder agents typically converge to their final evaluation before the third turn. Thus, three turns already enable meaningful argument exchange without increasing computation cost or producing redundant dialogue.
>
> To further validate MAJ-Eval's persona construction process, we added an ablation study using LLM-generated personas to engage in in-group debate.  The results are shown in the following table:
>
> | Method                                    | Overall Quality | Grammar Correctness | Answer Relevancy | Contextual Consistency | Educational Appropriateness |
> | ----------------------------------------- | --------------- | ------------------- | ---------------- | ---------------------- | --------------------------- |
> | LLM-Generated Persona (Claude-3.7-Sonnet) | 0.37            | **0.17**               | 0.39             | 0.20                   | 0.32                        |
> | MAJ-Eval (Claude-3.7-Sonnet)              | **0.47**            | 0.14                | **0.45**             | **0.33**             | **0.40**                  |
> | LLM-Generated Persona (Qwen-3-235B)       | 0.34            | 0.15                | 0.42             | 0.15                   | 0.20                        |
> | MAJ-Eval (Qwen-3-235B)                    | **0.43**        | **0.18**            | **0.43**             | **0.27**                   | **0.33**               |
>
> | Method                                    | Overall Quality | Fluency | PIO Consistency | Effect Direction | Evidence Strength |
> | ----------------------------------------- | --------------- | ------- | --------------- | ---------------- | ----------------- |
> | LLM-Generated Persona (Claude-3.7-Sonnet) | 0.32            | **0.14**    | 0.16            | 0.26             | 0.20              |
> | MAJ-Eval (Claude-3.7-Sonnet)              | **0.40**            | 0.10    | **0.21**            | **0.34**             | **0.28**              |
> | LLM-Generated Persona (Qwen-3-235B)       | 0.27            | 0.11    | 0.14            | 0.14             | **0.24**              |
> | MAJ-Eval (Qwen-3-235B)                    | **0.39**            | **0.34**    | **0.21**            | **0.26**             | 0.18              |
>
>
> As shown in the table, **MAJ-Eval consistently outperforms the baseline that relies on LLM-generated personas**, both in overall quality and in domain-specific dimensions such as Educational Appropriateness and Effect Direction. Qualitatively, we find that LLM-generated personas often fail to construct a comprehensive set of stakeholder groups. For example, for the medical-summarization task, Qwen generated only three stakeholder groups: Clinicians, Biomedical Researchers, and Healthcare Administrators/Policymakers. However, the actual stakeholders are more diverse, including Clinical Librarians, Public Health Consumers, etc. We will incorporate these additional baseline results and our expanded analysis into the revised version.
>
>
> **4. Cost and latency analysis**
>
> We acknowledge that MAJ-Eval is not inexpensive compared with single-shot  LLM prompts, but it is far more economical and convenient than hiring domain experts (see our cost analysis in Appendix A.5), especially when domain experts in specialized scenarios are hardly accessible (e.g., medical specialists). Based on our records, the total token usage per task is roughly 141,329 tokens. At Claude 3.7 Sonnet’s pricing of `$ 3` per million tokens, the cost is roughly `$ 0.42`, which is indeed marginal compared with the cost of recruiting human experts for evaluation. Regarding latency, MAJ-EVAL processes a single task in about 26.13 seconds on Qwen-3-235B and 34.20 seconds on Claude 3.7 Sonnet, which is faster than the human evaluation process. As automated metrics are insufficient and often not adopted in domain-specific evaluation tasks, researchers typically need to recruit expensive human experts for evaluation. We believe MAJ-Eval offers an efficient and human-aligned alternative, as its time and monetary cost are reasonable and are significantly more scalable than human expert evaluation.

---

> ### Author Response · Authors · 2025-12-04
> **Response to Your Feedback (Part 3 of 3)**
>
> **5. Generalizability to other domains**
>
> We would like to clarify that the primary contribution of MAJ-Eval lies in its ability to automatically generalize across specialized domains. The two specialized domains we chose (i.e., children’s education and medicine) represent popular research areas in NLP with sufficient available datasets. Also, these two domains totally differ in stakeholder perspectives, evaluation goals, and criteria. Therefore, MAJ-Eval’s consistent stronger performance on these two domains demonstrates its generalizability. We acknowledge that the availability of human-annotated data in other specialized domains is still limited, and we have noted this limitation in Section 8. In future work, we plan to incorporate additional tasks to further illustrate the generalizability of MAJ-Eval.
>
>
> **6. Clarification on rating scale**
>
> We would like to clarify that the rating scale in MAJ-Eval is fully customizable and can be adapted to different tasks or user preferences. In our experiments, we used a five-point Likert scale for the children’s storybook QAG task because this was the scale used by the dataset authors for human evaluation. For the medical summarization task, the original authors employed different scales across dimensions (e.g., 0–3, –1–2) and normalized them to 0–1. To give stakeholder agents finer granularity when assessing the content, we adopted a five-point Likert scale and then normalized the resulting scores.

---

### Official Review · Reviewer_v9ix · 2025-11-07

**Soundness:** 2
**Presentation:** 3
**Contribution:** 1
**Rating:** 2
**Confidence:** 4

**Summary:**

This paper addresses the challenge of evaluating complex, real-world NLP tasks, which often require aligning with diverse human stakeholder perspectives. The authors argue that existing "LLM-as-a-judge" methods are limited by arbitrarily designed agent personas and a lack of generalizability.

They propose MAJ-EVAL, a two-stage Multi-Agent-as-Judge evaluation framework. The first stage, "Stakeholder Persona Creation," aims to systematically ground agent personas by automatically extracting evaluative dimensions and perspectives from a provided set of domain-specific documents, such as research papers. The second stage, "Multi-Agent-as-Judge Debate Evaluation," instantiates LLM agents with these grounded personas and engages them in an in-group debate, allowing them to challenge and refine their initial judgments before a final score is aggregated.

**Strengths:**

The paper addresses a significant and practical problem: the need for scalable, multi-dimensional evaluation of NLP systems that aligns with diverse human stakeholders. The authors' motivation for moving beyond single-agent "LLM-as-a-judge" systems is well-articulated.

**Weaknesses:**

*Missing Comparison to LLM-Generated Personas*

This is the most signifiant issue. The paper's core claim is that its document-grounded personas are superior to "arbitrary" ones. However, the authors fail to test this against a strong, obvious baseline. The ablation study only compares their method to "simple role definition" (e.g., "You are a school teacher") , which is an insufficient comparison. A proper baseline would be to **prompt the LLM to generate a detailed, expert persona** directly from its own **pre-trained knowledge**, which assumes that the documents used to define the personas are already seen by the LLM. The paper provides no evidence that its complex, document-extraction pipeline is more effective than this simpler, more direct approach.

*Manual Document Selection Issue*

The entire "Stakeholder Persona Creation" stage —which is a core contribution—is highly dependent on the initial documents provided to the system. In Section 4.4, the authors state they manually selected "three representative documents" for one task and "two" for the other . This manual, subjective step introduces a major potential for bias. The framework is only "automatic" after this crucial manual selection.

*Instability of the Debate Mechanism*

The paper frames the debate's outcome as a success, but the data in Section 5.3.2 could be interpreted negatively. The authors admit that for some groups, like the "Language Researchers," the correlation with human ratings decreased after the debate. Their explanation is that the agents discussed new dimensions "beyond those used in human ratings". However, this shows the **debate is unstable and can cause agents to diverge** from the gold-standard evaluation criteria, which could be seen as a failure rather than a feature.

**Questions:**

See weakness.

---

> ### Author Response · Authors · 2025-12-04
> **Response to Your Feedback (Part 1 of 2)**
>
> Dear Reviewer, we sincerely appreciate your time and helpful feedback to make our paper stronger. In response, we would like to clarify and address the points as follows:
>
> **1. Missing Comparison to LLM-Generated Personas**
>
> We totally agree on conducting an ablation study using LLM-generated personas to engage in in-group debate and demonstrate the effectiveness of MAJ-Eval’s persona construction process. In response, we have conducted the ablation study with both Claude-3.7-Sonnet and Qwen-3. The results are shown in the following table:
>
> | Method                                    | Overall Quality | Grammar Correctness | Answer Relevancy | Contextual Consistency | Educational Appropriateness |
> | ----------------------------------------- | --------------- | ------------------- | ---------------- | ---------------------- | --------------------------- |
> | LLM-Generated Persona (Claude-3.7-Sonnet) | 0.37            | **0.17**               | 0.39             | 0.20                   | 0.32                        |
> | MAJ-Eval (Claude-3.7-Sonnet)              | **0.47**            | 0.14                | **0.45**             | **0.33**             | **0.40**                  |
> | LLM-Generated Persona (Qwen-3-235B)       | 0.34            | 0.15                | 0.42             | 0.15                   | 0.20                        |
> | MAJ-Eval (Qwen-3-235B)                    | **0.43**        | **0.18**            | **0.43**             | **0.27**                   | **0.33**               |
>
> | Method                                    | Overall Quality | Fluency | PIO Consistency | Effect Direction | Evidence Strength |
> | ----------------------------------------- | --------------- | ------- | --------------- | ---------------- | ----------------- |
> | LLM-Generated Persona (Claude-3.7-Sonnet) | 0.32            | **0.14**    | 0.16            | 0.26             | 0.20              |
> | MAJ-Eval (Claude-3.7-Sonnet)              | **0.40**            | 0.10    | **0.21**            | **0.34**             | **0.28**              |
> | LLM-Generated Persona (Qwen-3-235B)       | 0.27            | 0.11    | 0.14            | 0.14             | **0.24**              |
> | MAJ-Eval (Qwen-3-235B)                    | **0.39**            | **0.34**    | **0.21**            | **0.26**             | 0.18              |
>
>
> As shown in the table, **MAJ-Eval consistently outperforms the baseline that relies on LLM-generated personas**, both in overall quality and in domain-specific dimensions such as Educational Appropriateness and Effect Direction. Qualitatively, we find that LLM-generated personas often fail to construct a comprehensive set of stakeholder groups. For example, for the medical-summarization task, Qwen generated only three stakeholder groups: Clinicians, Biomedical Researchers, and Healthcare Administrators/Policymakers. However, the actual stakeholders are more diverse, including Clinical Librarians, Public Health Consumers, etc. We will incorporate these additional baseline results and our expanded analysis into the revised version.
>
> **2. Potential bias regarding manual document selection**
> We would like to clarify that the document selection step in MAJ-Eval is automated rather than fully manual. MAJ-Eval is designed as a generalizable framework to address the long-lasting challenge of recruiting qualified human experts in human evaluations. Accordingly, the document selection process is intended to represent the real-world expert recruitment procedure, in which different stakeholders are selected based on their domain-specific roles and expertise. For example, in educational content evaluation, teachers and educational experts are typically recruited to assess educational appropriateness.
>
> To ensure reproducibility and reduce subjectivity, we conducted a keyword-based search using Google Scholar. Based on our pilot test, we selected two to three documents per task because this number ensures stakeholder perspective diversity without being overly repetitive, which can make the personas less clear and focused. An example of our persona creation workflow, including the document selection procedure, is shown in Table 9 in the Appendix.
>
> We will revise section 4.4 to clarify the document selection process, inclusion criteria, and recommended document quantity to further improve transparency.

---

> ### Author Response · Authors · 2025-12-04
> **Response to Your Feedback (Part 2 of 2)**
>
> **3. Instability of the Debate Mechanism**
>
> We acknowledge that the debating mechanism does not always enhance the alignment. However, the debate mechanism allows agents to share the divergent perspectives according to the stakeholder roles they role-played with; thus, the evaluation will result in deeper considerations and more comprehensive evaluation perspectives (e.g., Educational Appropriateness from multiple stakeholders’ perspectives) compared with traditional surface-level dimensions (e.g., Grammar Correctness). This setup aligns with our goal of prioritizing multi-stakeholder-centered, task-relevant evaluation over linguistic similarity and quality.
>
> In addition, MAJ-Eval is designed to mitigate the inherent limitation of scoring-based evaluation in NLP research. In reality, different stakeholders have nuanced, complicated evaluation perspectives and rationales that cannot be fully captured and represented by fixed and often numerical metrics. Therefore, extensive research in CSCW and HCI has demonstrated that meaningful real-world assessment often requires user-centered interviews and qualitative analysis [1][2]. MAJ-Eval’s debate mechanism contributes to this direction by helping cover domain-specific, stakeholder-grounded dimensions that not only quantitatively align with but also qualitatively supplement existing human evaluation dimensions. Therefore, we do not consider the debate mechanism as ‘unstable’. Instead, we believe real-world evaluation should not be constrained by the current fixed human evaluation dimensions used in NLP research.
>
> [1] Khullar, A., Nalin, N., Prasad, A., Mampilli, A.J., & Kumar, N. (2025). Nurturing Capabilities: Unpacking the Gap in Human-Centered Evaluations of AI-Based Systems. Proceedings of the 2025 CHI Conference on Human Factors in Computing Systems.
> [2] Berman, Glen et al. “A Scoping Study of Evaluation Practices for Responsible AI Tools: Steps Towards Effectiveness Evaluations.” Proceedings of the 2024 CHI Conference on Human Factors in Computing Systems (2024): n. pag.

---

### Meta-Review · Area_Chair_fwah · 2026-01-07

**Summary:**

This paper proposes a multi-agent-as-judge framework that uses manually selected domain-specific documents to design detailed personas for domain stakeholders. This process improves performance over other evaluation methods and multi-agent evaluation using simpler as well as fully LLM generated personas. Allowing the generated personas to debate also improves alignment with human expert ratings.

The most important concerns that the reviewers raised are the following:

- Proper baselines
	- Although the authors added a new experiment using LLM generated personas (a clear baseline proposed by a reviewer), it is unclear whether the personas are truly comparable to the ones used in their method. Given that they show that simpler personas lead to worse performance, it is necessary to address this confounder in order to guarantee the conclusions provided in this work.

- Proper ablations on document selection
	- The authors refer to a “pilot test” when discussing how they generated the document selection process but this is not properly discussed in the paper. Given that the document selection step is the only manual step, it is necessary to analyze the stability of its performance with respect to different selected documents.

The reviewers also are skeptical about the generalizability of the paper’s conclusions based on the small number of datasets and models tested. Although I share this concern, the ones above are more important in my decision to reject this work.

**Reviewer Concerns:**

R1:
- LLM Generated Personas
	- The authors follow the reviewer’s guidance and provide an experiment that compares their method with purely LLM Generated Personas. However, the generated personas seem to be simpler than the ones generated by their method and the authors also showed (Appendix 3) that simpler personas degrade their performance. This is therefore an important confounder that has likely not been addressed.
- Manual Document Selection Issue
	- The authors claim that the document selection step is automated and point to Table 9 of the Appendix as confirmation. However, the only automated step in this process is the Google Scholar search process, leaving the query creation and document selection to the annotator. The reviewer’s concern that this process introduces bias that is unexplored in this work is not well-addressed.
- Debate Mechanism Instability
	- The authors acknowledge that the debate mechanism does not always improve alignment with human judgements but they believe that their debate mechanism “qualitatively supplement” existing human evaluation dimensions. This claim is substantiated only by the appearance of more evaluation dimensions in the “Language Researchers” domain after the debate but these dimensions were not rigorously evaluated. It is therefore unclear whether there are scenarios where the debate mechanism would be detrimental for the evaluation’s quality.

R2:
- Selected literature dependence per persona
	- Similar to the response to R1. The bias introduced through this manual document selection process is largely unexplored.
- Uncertainty analysis for correlation differences
	- They perform bootstrap sampling to show that the correlation with human judgements is statistically significant for most evaluation dimensions. As far as I understand, they do not seem to show whether the correlation improvements over previous methods such as G-Eval and ChatEval are statistically significant.
- Limited Judge backbones
-	- Too expensive to run other backbones.
- Shallow ablations (debate budget)
	- The authors claim that since the persona generation process is automated, they are unable to artificially change the persona counts or debate length and that the turn budget is already enough for convergence. Thus, they only add the ablation created for R1 on LLM generated personas. This seems reasonable but an ablation on the document selection step would be valuable.
- Cost and Latency tradeoffs
	- They provide approximate cost and latency for both model backbones and argue that they are much cheaper than a team of human experts. I would hope a more explicit version of this cost analysis (with respect to # of personas) was included in the paper.
- External validity still limited
	- They claim that the two datasets chosen are very different and their experiments therefore demonstrate generalizability. More datasets would likely improve the paper but it is not a non-negotiable.
- Rating Scale
	- The rating scales depend on each task individually.

R3:
- Persona validity
	- They automatically verify (with textual similarity) whether each persona is grounded in the source documents. Given that there are only a few personas per task and model, I would expect a more thorough qualitative evaluation of their quality.
- Where do their improvements come from?
	- They claim that comparing with the no-debate baseline shows that improvements come from the in-group debate. The “read-others-but-no-discussion” and self-consistency methods suggested by the reviewer were not tested.
- Ablations on cross-domain personas
	- Since evaluation metrics are different, this cannot be done. They provide the LLM generated persona ablation done for R1’s question.
- Demographic experiment
	- No extra evaluation on demographics was provided.
- Citations
	- The authors made no mention of the citations proposed by the reviewer.

R4:
- Persona extraction in under-documented domains
	- The authors claim that any documents can be used in under-documented domains. Since no exploration of varying documents is presented in this work, this claim is not well-substantiated.
- Agent convergence is not investigated
	- The authors claim that agents do not converge on a single correct answer but rather that each persona provides expert signals for specific evaluation dimensions.
- Expensive in practice
	- Same answer as cost question in R2.
- Limited domains and backbone models
	- Same as response to R2 addressing these concerns.
- Qualitative validation of the debate process is limited
	- Some qualitative analysis of the debate process is presented in Appendix A.6.
- Might be gaming the correlation with human scores instead of improving reasoning
	- The authors explain that bias introduced by human annotations are inevitable.

**Reviewer Scores:**

The reviewer scores would have likely remained the same based on outstanding concerns in each of the reviews.

---

### Decision · Program_Chairs · 2026-01-26

Reject